# From Cloze to Comprehension: Retrofitting Pre-trained Masked Language Models to Pre-trained Machine Reader[*]

**Weiwen Xu**[12][†]    **Xin Li**[2][‡]    **Wenxuan Zhang**[2]    **Meng Zhou**[3][†]
**Wai Lam**[1]    **Luo Si**[2]    **Lidong Bing**[2]
[1]The Chinese University of Hong Kong
[2]DAMO Academy, Alibaba Group
[3]Carnegie Mellon University
{wwxu,wlam}@se.cuhk.edu.hk   mengzhou@andrew.cmu.edu
{xinting.lx,saike.zwx,luo.si,l.bing}@alibaba-inc.com

## Abstract

We present Pre-trained Machine Reader (PMR), a novel method for retrofitting pre-trained masked language models (MLMs) to pre-trained machine reading comprehension (MRC) models without acquiring labeled data. PMR can resolve the discrepancy between model pre-training and downstream fine-tuning of existing MLMs. To build the proposed PMR, we constructed a large volume of general-purpose and high-quality MRC-style training data by using Wikipedia hyperlinks and designed a Wiki Anchor Extraction task to guide the MRC-style pre-training. Apart from its simplicity, PMR effectively solves extraction tasks, such as Extractive Question Answering and Named Entity Recognition. PMR shows tremendous improvements over existing approaches, especially in low-resource scenarios. When applied to the sequence classification task in the MRC formulation, PMR enables the extraction of high-quality rationales to explain the classification process, thereby providing greater prediction explainability. PMR also has the potential to serve as a unified model for tackling various extraction and classification tasks in the MRC formulation.[2]

## 1   Introduction

Span extraction, such as Extractive Question Answering (EQA) and Named Entity Recognition (NER), is a sub-topic of natural language understanding (NLU) with the goal of detecting token spans from the input text according to specific requirements like task labels or questions [45, 54]. Discriminative methods were used to execute such tasks and achieved state-of-the-art performance. As shown in the left part of Figure 1, these works tailored a task-specific fine-tuning head on top of pre-trained language models (PLMs) to perform sequence tagging or machine reading comprehension (MRC) [12, 36, 27]. The base PLMs are usually selected from pre-trained masked language models (MLM), such as RoBERTa [36] or BART [29] due to their comprehensive bi-directional modeling for the input text in the encoder. However, given the disparate nature of the learning objectives and different model architectures of MLM pre-training and task-specific fine-tuning, the discriminative

---

[*]This work was supported by Alibaba Group through Alibaba Research Intern Program. The work described in this paper was also partially supported by a grant from the Research Grant Council of the Hong Kong Special Administrative Region, China (Project Code: 14200620). [†] This work was done when Weiwen Xu and Meng Zhou interned at Alibaba DAMO Academy. [‡] Xin Li is the corresponding author.

[2]The code, data, and checkpoints are released at https://github.com/DAMO-NLP-SG/PMR

37th Conference on Neural Information Processing Systems (NeurIPS 2023).

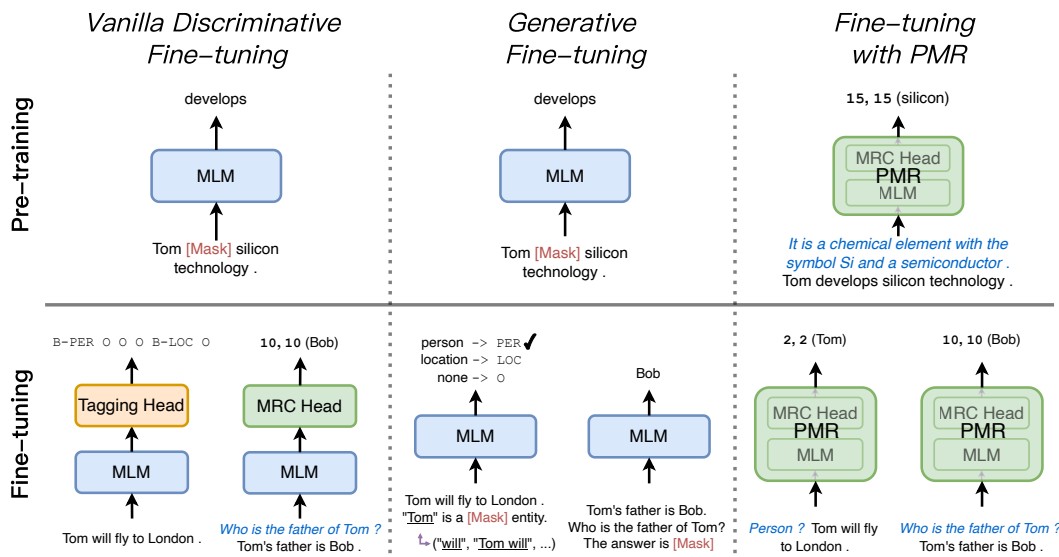

Figure 1: Comparison among three fine-tuning strategies for NER and EQA, namely, vanilla discriminative fine-tuning, generative fine-tuning, and fine-tuning by using the proposed PMR.

methods are less effective for adapting MLMs to downstream tasks when there is limited fine-tuning data available, leading to poor low-resource performance [6].

As shown in the middle part of Figure 1, generative fine-tuning is a popular solution to mitigate the gap between pre-training and fine-tuning [49, 50, 34]. This solution achieves remarkable few-shot performance in various span extraction tasks [10, 6, 38]. Specifically, generative methods formulate the downstream tasks as a language modeling problem in which PLMs generate response words for a given *prompt* (i.e., a task-specific template) as task prediction. Despite its success, tackling extraction tasks in a generative manner leads to several disadvantages. First, if it is used to generate the label token (e.g., "person" for PER entities) for a candidate span, the generative method needs to enumerate all possible span candidates to query PLMs [10]. This requirement can be computationally expensive for tasks with a long input text, such as EQA. Second, if the desired predictions are target spans (e.g., the "answer" in the EQA task), generative methods usually need to explore a large search space to generate span tokens. Moreover, it is also challenging to accurately generate structured outputs, e.g., the span-label pairs in the NER task, with PLMs originally trained on unstructured natural language texts. These limitations impede PLMs from effectively learning extraction patterns from increased volumes of training data. As a result, even instruction-tuned large language models like ChatGPT[3] are less effective than discriminative methods with smaller MLMs on extraction tasks [41, 43, 60, 31].

To bridge the gap between pre-training and fine-tuning without suffering from the aforementioned disadvantages, we propose a novel Pre-trained Machine Reader (PMR) as a retrofit of pre-trained MLM for more effective span extraction. As shown in the right part of Figure 1, PMR resembles common MRC models and introduces an MRC head on top of MLMs. But PMR is further enhanced by a comprehensive continual pre-training stage with large-scale MRC-style data. By maintaining the same MRC-style learning objective and model architecture as the continual pre-training during fine-tuning, PMR facilitates effective knowledge transfer in a discriminative manner and thus demonstrates great potential in both low-resource and rich-resource scenarios. Given that MRC has been proven as a universal paradigm [32, 33, 63, 23], our PMR can be directly applied to a broad range of span extraction tasks without additional task design.

To establish PMR, we constructed a large volume of general-purpose and high-quality MRC-style training data based on Wikipedia anchors (i.e., hyperlinked texts). As shown in Figure 2, for each Wikipedia anchor, we composed a pair of correlated articles. One side of the pair is the Wikipedia article that contains detailed descriptions of the hyperlinked entity, which we defined as the *definition*

---

[3]https://chat.openai.com

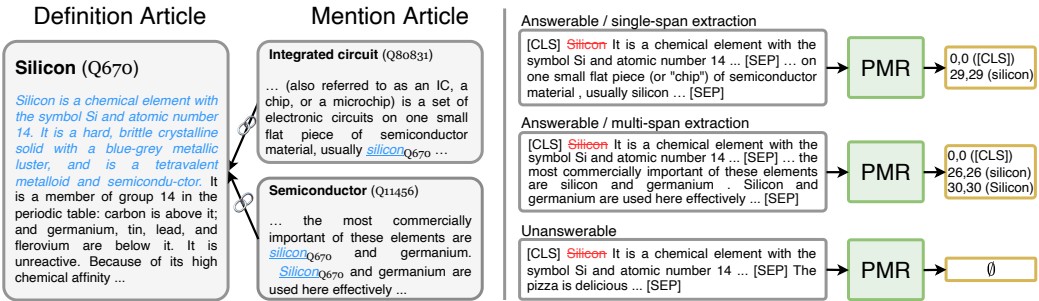

Figure 2: Construction of MRC-style data by using Wikipedia anchors.

*article*. The other side of the pair is the article that mentions the specific anchor text, which we defined as the *mention article*. We composed an MRC-style training instance in which the anchor is the answer, the surrounding passage of the anchor in the *mention article* is the context, and the definition of the anchor entity in the *definition article* is the query. Based on the above data, we then introduced a novel Wiki Anchor Extraction (WAE) problem as the pre-training task of PMR. In this task, PMR determines whether the context and the query are relevant. If so, PMR extracts the answer from the context that satisfies the query description.

We evaluated PMR on two representative span extraction tasks: NER and EQA. The results show that PMR consistently obtains better extraction performance compared with the vanilla MLM and surpasses the best baselines by large margins under almost all few-shot settings (up to 6.3 F1 on EQA and 16.3 F1 on NER). Additionally, we observe that sequence classification can be viewed as a special case of extraction tasks in our MRC formulation. In this scenario, it is surprising that PMR can identify high-quality rationale phrases from input text as the justifications for classification decisions. Furthermore, PMR has the potential to serve as a unified model for addressing various extraction and classification tasks in the MRC formulation.

In summary, our contributions are as follows. Firstly, we constructed a large volume of general-purpose and high-quality MRC-style training data to retrofit MLMs to PMRs. Secondly, by unifying pre-training and fine-tuning as the same discriminative MRC process, the proposed PMR obtains state-of-the-art results under all few-shot NER settings and three out of four few-shot EQA settings. Thirdly, with a unified MRC architecture for solving extraction and classification tasks, PMR also shows promising potential in explaining the sequence classification predictions and unifying NLU tasks.

## 2 PMR

This section describes PMR from the perspectives of model pre-training and downstream fine-tuning. For pre-training, we first introduce the proposed model with the training objective of WAE and then describe the curation procedure of WAE pre-training data from Wikipedia. For fine-tuning, we present how PMR can seamlessly be applied to various extraction tasks and solve them in a unified MRC paradigm.

### 2.1 Pre-training of PMR

PMR receives MRC-style data in the format of $(Q, C, \{A^k\}_{k=1}^K)$, where $Q$ is a natural language query and $C$ is the input context that contains the answers $\{A^k\}_{k=1}^K$ to the query. Each answer is a consecutive token span in the context, and zero ($K = 0$) or multiple ($K > 1$) answers may exist.

**Model Architecture.** PMR has two components: an MLM encoder and an extractor (Figure 3). The encoder receives the concatenation of query $Q$ and context $C$ as input $X$ and represents each input token as hidden states $H$.

$$X = [[\text{CLS}], Q, [\text{SEP}], C, [\text{SEP}]]$$
$$H = \textbf{MLM}(X) \in \mathbb{R}^{M \times d} \tag{1}$$

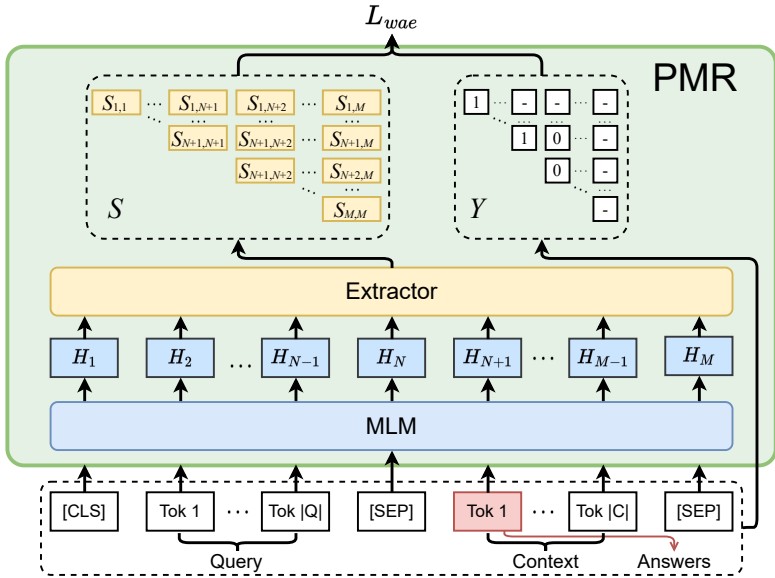

Figure 3: Model architecture of PMR. "-" indicates illegal candidate spans.

where [CLS] and [SEP] are special tokens inserted into the sequence, $M$ is the sequence length, and $d$ is the dimension of the hidden states. The encoder $\mathbf{MLM}(\cdot)$ denotes any pre-trained text encoder for retrofitting, e.g. RoBERTa.

The extractor receives the hidden states of any two tokens and predicts the probability score that tells if the span between the two tokens should be output as an answer. We applied the *general* way to compute the score matrix $S$ [39]:

$$S = \text{sigmoid}(\mathbf{FFN}(H)^T H) \in \mathbb{R}^{M \times M} \tag{2}$$

where $\mathbf{FFN}$ is the feed-forward network [57], and $S_{i,j}$ is the probability to extract the span $X_{i:j}$ as output. The *general* way avoids creating a large $\mathbb{R}^{M \times M \times 2d}$-shape tensor of the *concatenation* way [32], achieving higher training efficiency with fewer resources.

**Training Objective.** PMR is pre-trained with the WAE task, which checks the existence of answers in the context and extracts the answers if they exist. For the first goal, PMR determines whether the context contains spans that can answer the query:

$$L_{cls} = \mathbf{CE}(S_{1,1}, Y^{cls}) \tag{3}$$

where $\mathbf{CE}$ is the cross-entropy loss and $S_{1,1}$ at the [CLS] token denotes the query-context relevance score. If $Y^{cls} = 1$, the query and the context are relevant (i.e. answers exist). This task mimics the downstream situation in which there may be no span to be extracted in the context (e.g. NER) and encourages the model to learn through the semantic relevance of two pieces of text to recognize the unextractable examples.

Secondly, the model is expected to extract all correct spans from the context as answers, which can be implemented by predicting the answer positions:

$$L_{ext} = \sum_{N < i \leq j < M} \mathbf{CE}(S_{i,j}, Y_{i,j}^{ext}) \tag{4}$$

where $Y_{i,j}^{ext} = 1$ indicates that $X_{i:j}$ is an answer to $Q$, and $N$ is the positional offset of the context in $X$. Note that only $X_{i:j}$ with $N < i \leq j < M$ are legal answer span candidates (i.e., spans from the context). MRC-NER [32] predicted the start and end probabilities as two additional objectives. However, we find that these objectives are redundant for our matrix-based objective and incompatible with multi-span extraction.

The overall pre-training objective $L_{wae}$ is:

$$L_{wae} = L_{cls} + L_{ext} \tag{5}$$

**Data Preparation.** MLM training can be easily scaled to millions of raw texts with a self-supervised learning objective [12]. In contrast, training PMR in the MRC paradigm requires labeled triplets (query, context, and answers) as supervision signals, which is expensive to prepare for large-scale pre-training. To address this limitation, we automated the construction of general-purpose and high-quality MRC-style training data by using Wikipedia anchors.

As illustrated in Figure 2, a Wikipedia anchor hyperlinks two Wikipedia articles: the definition article that provides detailed descriptions of the anchor entity "Silicon", and the mention article where the anchor is mentioned. We leveraged the large scale of such hyperlink relations in Wikipedia as the distant supervision to automatically construct the MRC triplets. Specifically, we regarded an anchor as the MRC answer for the following context and query pair. The sentences surrounding the anchor in the mention article serve as the MRC context. The sentences from the first section of the definition article, which usually composes the most representative summary for the anchor entity [7], comprise the query. The query provides a precise definition of the anchor entity, and thus serves as a good guide for PMR to extract answers (i.e., anchor text) from the context.

Concretely, we considered sentences within a window size $W$ of the anchor as the MRC context and used the first $T$ sentences from the definition article as the query. Note that the context may cover multiple mentions of the same anchor entity. In this case, we treated all mentions as valid answers (i.e., $K > 1$) to avoid confusing the model training. More importantly, the preceding scenario naturally resembles multi-span extraction tasks like NER. To prevent information leakage, we anonymized the anchor entity in the query by using "it" to substitute text spans that overlapped more than 50% with the anchor entity name. we did not use the "[MASK]" token because it does not exist in the data of downstream tasks.

In addition to the above answerable query and context pairs prepared through hyperlink relation, we introduced unanswerable examples by pairing a context with an irrelevant query (i.e., query and context pairs without the hyperlink association). The unanswerable examples are designed to help the model learn the ability to identify passage-level relevance and avoid extracting any answer (i.e., $K = 0$) for such examples.

## 2.2 Fine-tuning PMR for Extraction Tasks

We unified downstream span extraction tasks in our MRC formulation, which typically falls into two categories: (1) span extraction with pre-defined labels (e.g., NER) in which each task label is treated as a query to search the corresponding answers in the input text (context) and (2) span extraction with natural questions (e.g., EQA) in which the question is treated as the query for answer extraction from the given passage (context). Then, in the output space, we tackled span extraction problems by predicting the probability $S_{i,j}$ of context span $X_{i:j}$ being the answer. The detailed formulation and examples are provided in Appendix A.2.

# 3 Experimental Setup

**Implementation.** We used the definition articles of the entities that appear as anchors in at least 10 other articles to construct the query. As mentioned in Sec. 2, we prepared 10 answerable query and context pairs for each anchor entity. Then, we paired the query with 10 irrelevant contexts to formulate unanswerable MRC examples. The resulting pre-training corpus consists of 18 million MRC examples (6.4 billion words). We also tried various advanced data construction strategies, such as relevance-driven and diversity-driven ones, to construct query and context pairs. However, no significant performance gain is observed. A detailed comparison is provided in Appendix A.4.

The encoder of PMR is initialized with RoBERTa, a popular MLM with competitive downstream performance. The extractor is randomly initialized, introducing additional 1.6M parameters. In terms of the pre-training efficiency, with four A100 GPUs, it only takes 36 and 89 hours to complete 3-epoch training of PMR for base-sized and large-sized models, respectively. Additional data preprocessing details and hyper-parameter settings can be found in Appendix A.3.

**Downstream Extraction Tasks.** We evaluated two extraction tasks: EQA and NER.

**EQA:** We evaluated PMR on MRQA benchmark [15]. For the few-shot setting, we used the few-shot MRQA datasets sampled by Splinter [46]. Although **BioASQ** and **TbQA** are originally used for OOD

| Model | Size | Unified | Shot | SQuAD | TriviaQA | NQ | NewsQA | SearchQA | HotpotQA | BioASQ | TbQA | Avg. |
|---|---|---|---|---|---|---|---|---|---|---|---|---|
| RoBERTa | 125M | ✗ | 16-shot | $7.7_{4.3}$ | $7.5_{4.4}$ | $17.3_{3.3}$ | $1.4_{0.8}$ | $6.9_{2.7}$ | $10.5_{2.5}$ | $16.7_{7.1}$ | $3.3_{2.1}$ | 8.9 |
| RBT-Post | 125M | ✗ | | $8.8_{1.2}$ | $11.9_{3.4}$ | $15.2_{5.6}$ | $2.6_{1.2}$ | $12.4_{2.5}$ | $7.0_{3.5}$ | $22.8_{5.1}$ | $5.3_{1.5}$ | 10.8 |
| SpanBERT | 110M | ✗ | | $18.2_{6.7}$ | $11.6_{2.1}$ | $19.6_{3.0}$ | $7.6_{4.1}$ | $13.3_{6.0}$ | $12.5_{5.5}$ | $15.9_{4.4}$ | $7.5_{2.9}$ | 13.3 |
| Splinter | 125M | ✓ | | $54.6_{6.4}$ | $18.9_{4.1}$ | $27.4_{4.6}$ | $20.8_{2.7}$ | $26.3_{3.9}$ | $24.0_{5.0}$ | $28.2_{4.9}$ | $19.4_{4.6}$ | 27.5 |
| FewshotBART | 139M | ✓ | | $\mathbf{55.5}_{2.0}$ | $\mathbf{50.5}_{1.0}$ | $\mathbf{46.7}_{2.3}$ | $\mathbf{38.9}_{0.7}$ | $39.8_{0.04}$ | $\mathbf{45.1}_{1.5}$ | $\mathbf{49.4}_{0.02}$ | $19.9_{1.3}$ | $\mathbf{43.2}$ |
| PMR$_{base}$ | 125M | ✓ | | $46.5_{4.0}$ | $47.7_{3.7}$ | $32.6_{3.0}$ | $26.2_{2.7}$ | $\mathbf{50.1}_{3.2}$ | $32.9_{2.9}$ | $49.1_{4.1}$ | $\mathbf{27.9}_{3.3}$ | 39.1 |
| PMR$_{large}$ | 355M | ✓ | | $60.3_{4.0}$ | $56.2_{3.1}$ | $43.6_{1.7}$ | $30.1_{3.7}$ | $58.2_{5.0}$ | $46.1_{4.7}$ | $54.2_{3.4}$ | $31.0_{1.8}$ | 47.5 |
| RoBERTa | 125M | ✗ | 32-shot | $18.2_{5.1}$ | $10.5_{1.8}$ | $22.9_{0.7}$ | $3.2_{1.7}$ | $13.5_{1.8}$ | $10.4_{1.9}$ | $23.3_{6.6}$ | $4.3_{0.9}$ | 13.3 |
| RBT-Post | 125M | ✗ | | $12.7_{3.7}$ | $13.4_{2.5}$ | $20.7_{3.0}$ | $2.9_{0.6}$ | $13.4_{2.5}$ | $10.8_{0.8}$ | $26.0_{5.2}$ | $5.3_{1.2}$ | 13.2 |
| SpanBERT | 110M | ✗ | | $25.6_{7.7}$ | $15.1_{6.4}$ | $25.1_{1.6}$ | $7.2_{4.6}$ | $14.6_{8.5}$ | $13.2_{3.5}$ | $25.1_{3.3}$ | $7.6_{2.3}$ | 16.7 |
| Splinter | 125M | ✓ | | $59.2_{2.1}$ | $28.9_{3.1}$ | $33.6_{2.4}$ | $27.5_{3.2}$ | $34.8_{1.8}$ | $34.7_{3.9}$ | $36.5_{3.2}$ | $27.6_{4.3}$ | 35.4 |
| FewshotBART | 139M | ✓ | | $56.8_{2.1}$ | $52.5_{0.7}$ | $50.1_{1.1}$ | $40.4_{1.5}$ | $41.8_{0.02}$ | $47.9_{1.4}$ | $52.3_{0.02}$ | $22.7_{2.3}$ | 45.6 |
| PMR$_{base}$ | 125M | ✓ | | $\mathbf{61.0}_{2.9}$ | $\mathbf{55.6}_{2.7}$ | $41.6_{3.6}$ | $31.2_{2.7}$ | $\mathbf{58.0}_{2.8}$ | $43.5_{0.8}$ | $\mathbf{58.9}_{3.1}$ | $\mathbf{35.1}_{3.5}$ | $\mathbf{48.1}$ |
| PMR$_{large}$ | 355M | ✓ | | $70.0_{3.2}$ | $66.3_{2.5}$ | $48.5_{3.5}$ | $36.6_{2.1}$ | $64.8_{2.2}$ | $52.9_{2.5}$ | $62.9_{2.4}$ | $36.4_{3.2}$ | 54.8 |
| RoBERTa | 125M | ✗ | 128-shot | $43.0_{7.1}$ | $19.1_{2.9}$ | $30.1_{1.9}$ | $16.7_{3.8}$ | $27.8_{2.5}$ | $27.3_{3.9}$ | $46.1_{1.4}$ | $8.2_{1.1}$ | 27.3 |
| RBT-Post | 125M | ✗ | | $11.0_{1.0}$ | $26.9_{0.9}$ | $24.6_{4.9}$ | $4.8_{1.3}$ | $27.8_{1.9}$ | $13.5_{1.8}$ | $30.2_{1.6}$ | $8.9_{1.3}$ | 18.5 |
| SpanBERT | 110M | ✗ | | $55.8_{3.7}$ | $26.3_{2.1}$ | $36.0_{1.9}$ | $29.5_{7.3}$ | $26.3_{4.3}$ | $36.6_{3.4}$ | $52.2_{3.2}$ | $20.9_{5.1}$ | 35.5 |
| Splinter | 125M | ✓ | | $72.7_{1.0}$ | $44.7_{3.9}$ | $46.3_{0.8}$ | $43.5_{1.3}$ | $47.2_{4.5}$ | $54.7_{1.4}$ | $63.2_{4.1}$ | $42.6_{2.5}$ | 51.9 |
| FewshotBART | 139M | ✓ | | $68.0_{0.3}$ | $50.1_{1.8}$ | $\mathbf{53.9}_{0.9}$ | $47.9_{1.2}$ | $58.1_{1.4}$ | $54.8_{0.8}$ | $68.5_{1.0}$ | $29.7_{2.4}$ | 53.9 |
| PMR$_{base}$ | 125M | ✓ | | $\mathbf{73.1}_{0.9}$ | $\mathbf{63.6}_{1.9}$ | $51.9_{1.7}$ | $46.9_{0.6}$ | $\mathbf{67.5}_{1.2}$ | $\mathbf{56.4}_{1.5}$ | $\mathbf{79.2}_{1.3}$ | $\mathbf{42.7}_{2.2}$ | $\mathbf{60.2}$ |
| PMR$_{large}$ | 355M | ✓ | | $81.7_{1.2}$ | $70.3_{0.5}$ | $57.4_{2.6}$ | $52.3_{1.4}$ | $70.0_{1.1}$ | $65.9_{1.0}$ | $78.8_{0.5}$ | $45.1_{1.2}$ | 65.2 |
| RoBERTa | 125M | ✗ | 1024-shot | $73.8_{0.8}$ | $46.8_{0.9}$ | $54.2_{1.1}$ | $47.5_{1.1}$ | $54.3_{1.2}$ | $61.8_{1.3}$ | $84.1_{1.1}$ | $35.8_{2.0}$ | 57.3 |
| RBT-Post | 125M | ✗ | | $73.0_{0.4}$ | $49.9_{1.2}$ | $48.1_{2.6}$ | $46.8_{0.9}$ | $54.8_{1.0}$ | $59.7_{0.8}$ | $86.4_{0.3}$ | $37.3_{1.3}$ | 57.0 |
| SpanBERT | 110M | ✗ | | $77.8_{0.9}$ | $50.3_{4.0}$ | $57.5_{0.9}$ | $49.3_{2.0}$ | $60.1_{2.2}$ | $67.4_{1.6}$ | $89.3_{0.6}$ | $42.3_{1.9}$ | 61.8 |
| Splinter | 125M | ✓ | | $82.8_{0.8}$ | $64.8_{0.9}$ | $65.5_{0.5}$ | $57.3_{0.8}$ | $67.3_{1.3}$ | $\mathbf{70.3}_{0.8}$ | $91.0_{1.0}$ | $54.5_{1.5}$ | 69.2 |
| FewshotBART | 139M | ✓ | | $76.7_{0.8}$ | $52.8_{3.2}$ | $58.7_{1.4}$ | $56.8_{1.6}$ | $69.5_{1.0}$ | $63.1_{0.8}$ | $91.3_{0.5}$ | $47.9_{1.9}$ | 64.6 |
| PMR$_{base}$ | 125M | ✓ | | $\mathbf{82.8}_{0.2}$ | $\mathbf{69.5}_{0.6}$ | $\mathbf{66.6}_{1.1}$ | $\mathbf{59.2}_{0.3}$ | $\mathbf{74.6}_{0.6}$ | $69.9_{0.3}$ | $\mathbf{94.4}_{0.5}$ | $\mathbf{54.5}_{1.0}$ | $\mathbf{71.4}$ |
| PMR$_{large}$ | 355M | ✓ | | $87.6_{0.7}$ | $73.7_{0.8}$ | $71.8_{1.2}$ | $64.4_{0.9}$ | $76.0_{0.9}$ | $74.7_{0.9}$ | $94.7_{0.2}$ | $60.8_{1.6}$ | 75.5 |

Table 1: EQA results (F1) in four few-shot settings. In each setting, we reported the mean and standard deviation over five splits of training data. The results of PMR$_{large}$ are written in blue for demonstration purposes because it has more parameters than others (all base-sized.) The **Size** column indicates the parameter size of models. The **Unified** column indicates if the model bridges the gaps between pre-training and fine-tuning (✓) or not (✗). The results of the best base-sized models are written in bold.

evaluation, Splinter [46] constructed few-shot training sets for both datasets by sampling examples from their original dev sets. We followed FewshotQA [6] to build the dev set that has the same size as the training set for model selection. For the full-resource experiments, we followed MRQA [15] in both in-domain and OOD evaluations.

**NER:** We evaluated PMR on two flat NER datasets, namely, CoNLL and WNUT, and two nested NER datasets, namely, ACE04 and ACE05 [54, 11, 42, 58]. For the few-shot setting, we constructed five splits of K-shot training data, where K∈{4,8,32,64} sentences are sampled for each tag [38]. We also constructed the dev set that has the same size as the training set.

**Baselines.** We compared PMR with (1) vanilla MLMs: RoBERTa [36] and SpanBERT [20] with a randomly-initialized extractor, (2) RBT-Post, a continually pre-trained RoBERTa using our Wikipedia data but with the MLM objective; and (3) models bridging the gaps between pre-training and fine-tuning, namely, Splinter [46], FewshotBART [6], EntLM [40], T5-v1.1 [44], and UIE [38], where the latter four are generative methods.

## 4 Main Results

**Few-shot Results.** The few-shot results of EQA and NER are presented in Tables 1 and 2. The poor few-shot capability of RoBERTa suggests that fine-tuning an MRC extractor from scratch with limited data is extremely challenging. Our PMR achieves notably better results on all few-shot levels of the two tasks than RoBERTa, obtaining an average improvement of 34.8 F1 and 18.6 F1 on 32-shot EQA and 4-shot NER, respectively. Other models that bridge the gaps between pre-training and fine-tuning also perform much better than RoBERTa and RBT-Post, which is consistent with our findings. For EQA, although FewshotBART benefits from a larger output space of the generative

| Model | Size | Unified | | CoNLL | WNUT | ACE04 | ACE05 | Avg. | | CoNLL | WNUT | ACE04 | ACE05 | Avg. |
|---|---|---|---|---|---|---|---|---|---|---|---|---|---|---|
| | | | | *4-shot* | | | | | *8-shot* | | | | | |
| RoBERTa | 125M | ✗ | | $32.4_{7.2}$ | $29.7_{3.8}$ | $47.5_{4.5}$ | $48.1_{3.7}$ | 39.4 | | $47.2_{7.4}$ | $35.1_{2.0}$ | $63.7_{2.4}$ | $58.8_{2.7}$ | 51.2 |
| RBT-Post | 125M | ✗ | | $31.8_{7.8}$ | $28.7_{3.6}$ | $48.7_{4.3}$ | $44.3_{4.7}$ | 38.4 | | $43.7_{7.1}$ | $35.0_{4.8}$ | $61.8_{2.2}$ | $58.3_{2.6}$ | 49.7 |
| EntLM | 125M | ✗ | | $\mathbf{67.6_{2.2}}$ | $27.2_{1.8}$ | - | - | - | | $71.3_{0.7}$ | $33.3_{0.8}$ | - | - | - |
| UIE | 220M | ✓ | | $52.0_{3.3}$ | $28.3_{3.1}$ | $45.2_{2.3}$ | $41.3_{3.0}$ | 41.7 | | $66.5_{2.0}$ | $39.3_{1.6}$ | $52.4_{1.8}$ | $51.8_{1.2}$ | 52.5 |
| PMR$_\text{base}$ | 125M | ✓ | | $65.1_{4.2}$ | $\mathbf{40.8_{3.1}}$ | $\mathbf{65.3_{2.5}}$ | $\mathbf{60.7_{2.9}}$ | **58.0** | | $\mathbf{73.9_{3.2}}$ | $\mathbf{41.1_{3.9}}$ | $\mathbf{70.7_{1.8}}$ | $\mathbf{68.0_{1.3}}$ | **63.4** |
| PMR$_\text{large}$ | 355M | ✓ | | $65.7_{4.5}$ | $40.5_{3.3}$ | $65.2_{3.5}$ | $66.1_{2.8}$ | 59.4 | | $70.3_{4.4}$ | $46.4_{2.5}$ | $71.7_{1.5}$ | $69.9_{2.1}$ | 64.6 |
| | | | | *32-shot* | | | | | *64-shot* | | | | | |
| RoBERTa | 125M | ✗ | | $77.8_{1.8}$ | $47.9_{1.6}$ | $76.8_{0.3}$ | $74.4_{0.6}$ | 69.2 | | $80.8_{1.1}$ | $50.6_{1.1}$ | $78.7_{1.3}$ | $77.9_{0.6}$ | 72.0 |
| RBT-Post | 125M | ✗ | | $77.1_{0.7}$ | $45.8_{1.9}$ | $75.7_{0.7}$ | $73.6_{0.9}$ | 68.1 | | $80.9_{1.0}$ | $49.8_{2.8}$ | $79.2_{1.4}$ | $77.5_{1.1}$ | 71.9 |
| EntLM | 125M | ✗ | | $78.9_{0.9}$ | $42.4_{0.9}$ | - | - | - | | $82.1_{1.1}$ | $46.2_{1.1}$ | - | - | - |
| UIE | 220M | ✓ | | $79.6_{1.1}$ | $46.2_{1.2}$ | $68.0_{0.5}$ | $66.3_{0.8}$ | 65.0 | | $83.2_{0.8}$ | $48.4_{1.2}$ | $74.7_{0.6}$ | $72.2_{0.4}$ | 69.6 |
| PMR$_\text{base}$ | 125M | ✓ | | $\mathbf{81.7_{1.2}}$ | $\mathbf{50.3_{1.4}}$ | $\mathbf{79.0_{1.1}}$ | $\mathbf{76.9_{1.3}}$ | **72.0** | | $\mathbf{84.4_{0.9}}$ | $\mathbf{51.5_{2.2}}$ | $\mathbf{81.6_{0.8}}$ | $\mathbf{79.5_{0.5}}$ | **74.3** |
| PMR$_\text{large}$ | 355M | ✓ | | $83.2_{0.6}$ | $50.4_{1.6}$ | $79.8_{1.2}$ | $77.2_{1.5}$ | 72.7 | | $84.5_{1.7}$ | $53.1_{2.8}$ | $81.2_{1.2}$ | $79.7_{1.1}$ | 74.6 |

Table 2: NER results (F1) in four few-shot settings. EntLM is not applicable for nested NER tasks.

| | Size | SQuAD | BioASQ | DROP | DuoRC | RACE | RE | TbQA | Avg. |
|---|---|---|---|---|---|---|---|---|---|
| T5-v1.1 | 800M | 93.9 | **72.8** | 47.3 | 63.9 | **57.5** | 87.1 | **61.5** | 65.0 |
| RoBERTa | 355M | 94.2 | 65.8 | 54.8 | 58.6 | 49.0 | 88.1 | 54.7 | 61.8 |
| PMR | 355M | **94.5** | 71.4 | **62.7** | **64.1** | 53.6 | **88.2** | 57.5 | **66.3** |

Table 3: Performance on OOD EQA. We used the SQuAD training data to train the models and evaluate them on MRQA OOD dev sets.

model to achieve better transferability in an extremely low-resource EQA setting (i.e., 16 shot), processing such large output is far more complicated than the discriminative MRC and thus is prone to overfitting. MRC-based PMR demonstrates higher effectiveness in learning extraction capability from more training examples than FewshotBART, consequently yielding better performance on EQA when at least 32 examples are provided. Note that the compared baselines for EQA and NER are slightly different due to their different applicability. For example, UIE mainly emphasizes a structured prediction and is not applicable to complicated extraction tasks like EQA. EntLM, which aims to generate label tokens, is also not applicable to EQA. The findings further reveal that PMR can work reasonably well as a zero-shot learner (Appendix A.5).

**OOD Generalization.** Domain generalization is another common low-resource scenario in which the knowledge can be transferred from a resource-rich domain to a resource-poor domain. We evaluated the domain generalization capability of the proposed PMR on the MRQA benchmark. The MRQA benchmark provides meaningful OOD datasets that are mostly converted from other tasks (e.g., multi-choice QA) and substantially differ from SQuAD in terms of text domain.

Table 3 shows that PMR significantly surpasses RoBERTa on all six OOD datasets (+4.5 F1 on average), although they have similar in-domain performance on SQuAD. This finding verifies that our PMR with MRC-style pre-training can help capture QA patterns that are more generalizable to unseen domains. In addition, PMR with less than half the parameters of T5-v1.1, achieves a better generalization capability.

**Full-resource Results.** Although using the full-resource training data can alleviate the pretraining-finetuning discrepancy, MRC-style continual pre-training still delivers reasonable performance gains. As shown in Table 4, PMR achieves 0.9 and 1.5 F1 improvements over RoBERTa on EQA and NER, respectively. Further analysis shows that PMR can do better at comprehending the input text (Appendix 5.3). We also explore the upper limits of PMR by employing a larger and stronger MLM, i.e. ALBERT$_\text{xxlarge}$ [27], as the backbone. The results show additional improvements of our PMR over ALBERT$_\text{xxlarge}$ on EQA (Appendix A.6).

| Model | Size | Unified | EQA | NER |
|---|---|---|---|---|
| RBT-Post | 355M | ✗ | 81.9 | 79.8 |
| SpanBERT | 336M | ✗ | 81.7 | 77.3 |
| T5-v1.1 | 800M | ✓ | 82.0 | 76.0 |
| UIE | 800M | ✓ | - | 79.6 |
| RoBERTa | 355M | ✗ | 84.0 | 80.8 |
| PMR | 355M | ✓ | **84.9** | **82.3** |

Table 4: Full-resource results on EQA and NER. For EQA, we reported the average F1 score on six MRQA in-domain dev sets. For NER, we used four datasets.

| | Size | RACE | DREAM | MCTest | SST-2 | MNLI | Avg. |
|---|---|---|---|---|---|---|---|
| T5-v1.1 | 800M | 81.7 | 75.7 | 86.7 | **96.7** | **90.4** | 86.2 |
| RBT-Post | 355M | 80.6 | 81.3 | 86.9 | 96.1 | 89.7 | 86.9 |
| RoBERTa | 355M | 83.2 | **84.2** | 89.3 | 96.4 | 90.1 | 88.6 |
| PMR | 355M | 82.8 | 83.8 | **92.3** | 96.5 | 89.9 | **89.1** |

Table 5: Full-resource performance (Acc.) on sequence classification tasks.

| Input and extracted rationales | Label |
|---|---|
| It's all pretty **cynical and condescending**, too. | Negative |
| Perhaps the **heaviest, most joyless movie** ever made about giant dragons taking over the world. | Negative |
| This is the **best American movie** about troubled teens since 1998's whatever. | Positive |
| An **experience so engrossing** it is like being buried in a new environment. | Positive |

Table 6: Case study on SST-2. PMR can additionally extract token-level rationales (in red and bold) to support the sequence classification. A total of 224 rationales out of 300 (74%) are reasonable.

## 5 Discussions

### 5.1 Explainable Sequence Classification

**Settings.** The relevance classification between the query and the context in our WAE objective can be inherited to tackle downstream sequence classification problems, another important topic of NLU. To demonstrate, we considered two major sequence classification tasks in our MRC formulation. The first is sequence classification with pre-defined task labels, such as sentiment analysis. Each task label is used as a query for the input text (i.e. the context in PMR)[4]; The second is sequence classification with natural questions on multiple choices, such as multi-choice QA (MCQA). We treated the concatenation of the question and one choice as the query for the given passage (i.e., the context). In the output space, these problems are tackled by conducting relevance classification on $S_{1,1}$ (extracting [CLS] if relevant). Examples are included in Appendix A.2. We tested PMR on (1) three MCQA tasks: DREAM [53], RACE [26], and MCTest [47]; (2) a sentence-pair classification task: MNLI [61]; and (3) a sentence classification task: SST-2 [52].

**Results.** Table 5 presents the results of PMR and previous strong baselines on the above-mentioned tasks. Compared with RoBERTa, PMR exhibits comparable results on all tasks and achieved slightly better average performance. This indicates that solving sequence classification in an MRC manner is feasible and effective.

**Explainability.** In sequence classification, we restricted the extraction space to the [CLS] token and used the extraction probability of this token to determine which class label (i.e. the query) the input text (i.e. the context) corresponds to. Note that while predicting the extraction probability of [CLS], PMR also calculates the extraction scores of context spans, just as done for the span extraction tasks. Therefore, we leveraged the fine-tuned PMR to additionally extract a span with the highest $S_{i,j}$ score from the input of SST-2, as exemplified in Table 6. The extracted spans are clear rationales that support the sequence-level prediction (i.e., the overall sentiment of the input sentence). To verify this, we conducted a quantitative analysis by randomly checking 300 test instances. The results show that approximately 74% of the extracted spans are reasonable rationales for sentiment prediction. These findings suggest that the fine-tuned PMR on SST-2 effectively preserves the conditional span extraction capability inherited from the MRC pre-training. Such a capability shows that the model captures the semantic interactions between relevant context spans and the label-encoded query (e.g., "Negative. Feeling bad."). In addition to the performance improvement, PMR provides high explainability for the classification results.

---

[4]Sentence-pair classification is classified into this type, where the concatenation of two sentences denotes the context.

## 5.2 Unifying Extraction and Classification with PMR

In the previous sections, we demonstrate that various extraction and classification tasks can be separately tackled in the same MRC formulation. We further explore the potential that fine-tuning a unified model for solving multiple tasks of different types.

**Settings.** We use two datasets, one from CoNLL NER (of extraction type) and the other from DREAM (of classification type), to train a multi-task model. For evaluation, we conduct three groups of experiments, where the models are evaluated on (1) Held-in: testing sets from training tasks, (2) Held-out Datasets: testing sets from other tasks of the same type with training tasks, and (3) Held-out Tasks: testing sets from unseen tasks

**Results.** As shown in Figure 4, the held-in results show that the multi-task fine-tuned RoBERTa suffers from a significant performance drop on DREAM compared to the single-task fine-tuned RoBERTa. This indicates that multi-task learning is difficult for a discriminative model if the task head is not well-trained. In contrast, the multi-task PMR is on par with PMR on CoNLL and slightly underperforms PMR on DREAM. Such a finding suggests that the MRC-style pre-training enhances PMR's capability to learn extraction and classification patterns from downstream NLU tasks, enabling PMR to serve as a unified model for solving various NLU tasks. Though generative models like T5-v1.1 (800M) could also unify NLU tasks through a conditional generation manner [44, 38], the overall performance,

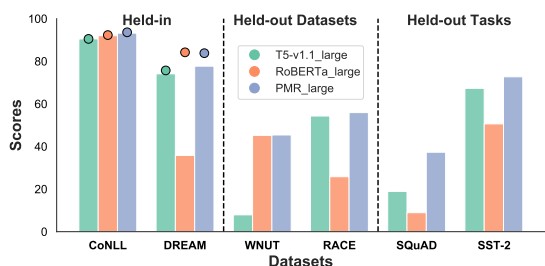

Figure 4: The results of Held-in, Held-out Datasets, and Held-out Tasks evaluation. The six data points in the Held-in group denote the single-task fine-tuned performance of three models on two tasks respectively.

especially the held-out performance, is lower than the smaller-sized PMR (355M). This suggests that the discriminative PMR may be better than generative models at unifying NLU tasks.

## 5.3 Better Comprehending capability

To verify that PMR can better comprehend the input text, we feed the models with five different query variants during CoNLL evaluation. The five variants are:

- Defaulted query:
  `"[Label]". [Label description]`
- Query template (v1):
  `What is the "[Label]" entity, where [Label description]?`
- Query template (v2):
  `Identify the spans (if any) related to "[Label]" entity. Details: [Label description]`
- Paraphrasing label description with ChatGPT (v1):
  `"[Label]". [Paraphrased Label description v1]`
- Paraphrasing label description with ChatGPT (v1):
  `"[Label]". [Paraphrased Label description v2]`

In Figure 5, we show the statistic results of the three models on CoNLL when five different query templates are used respectively during evaluation. Among the models, PMR demonstrated significantly higher and more stable performance than RoBERTa and T5-v1.1. Such a finding verifies our assumption that PMR can effectively comprehend the latent semantics of the input text despite being rephrased with varying lexical usage from the default query used for fine-tuning models.

## 6 Related Work

**Gaps between Pre-training and Fine-tuning.** The prevailing approaches tackle span extraction tasks in a discriminative manner with tailored task-specific classifiers (e.g. tagging or MRC head)

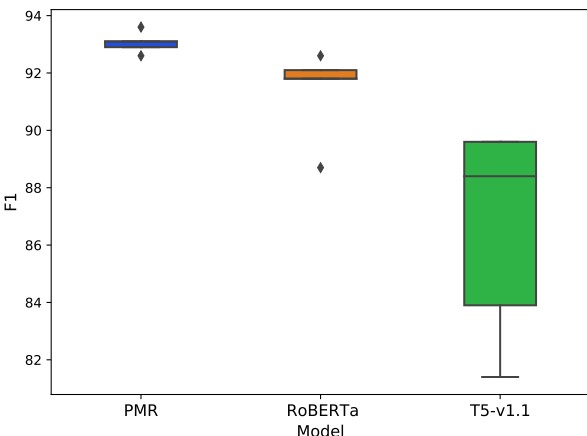

Figure 5: CoNLL performance when the models are fed with five different templates respectively during evaluation.

[19, 9, 12]. Recent research reported that generative fine-tuning methods can effectively bridge the gaps between pre-training and fine-tuning and achieve remarkable few-shot NLU performance by representing downstream NLU tasks as the same language model pre-training problem [49, 50, 34, 16, 35]. In the scope of span extraction, these works can be classified into three categories: generating label tokens based on the prompt span [10, 40], generating span tokens based on the prompt label [6, 38], and directly generating label-span pairs with soft prompt [8]. Another way to mitigate the gap is to adapt MLMs into an appropriate paradigm for solving span extraction. Typically, this can be achieved through the use of task-specific data from similar tasks, referred to as "pre-finetuning" [24, 1, 68]. However, these methods may not be effective in domain-specific tasks or for non-English languages due to the lack of labeled data. Several studies leveraged abundant raw text to construct MRC examples for retrofitting MLMs [30, 18, 46]. By contrast, PMR employs ubiquitous hyperlink information to construct MRC data, which guarantees highly precise MRC triplets. In addition, PMR also provides a unified model for both sequence classification and span extraction, thereby enabling strong explainability through the extraction of high-quality rationale phrases.

**Hyperlinks for NLP.** Hyperlinks are utilized in two ways. First, hyperlinks can be regarded as a type of relevance indicator in model pre-training [67], passage retrieval [7, 51], and multi-hop reasoning [2, 66]. Second, the anchors labeled by hyperlinks can serve as entity annotations for representation learning [64, 5]. PMR is the first one to combine the advantages of both scenarios. In this work, we paired MRC query and context based on the relevance of hyperlinks and automatically labeled the anchors as MRC answers.

## 7 Conclusions

This work presents a novel MRC-style pre-training model called PMR. PMR can fully resolve the learning objective and model architecture gaps that frequently appear in fine-tuning existing MLMs. Experimental results from multiple dimensions, including effectiveness in solving few-shot tasks and OOD generalization, show the benefits of bridging the gap between pre-training and fine-tuning for span extraction tasks. PMR also shows promising potential in explaining the sequence classification process and unifying NLU tasks.

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

# A   Appendix

## A.1   Limitations

**Multilinguality**   Although constructing large-scale MRC-style training data is feasible for resource-rich languages, such as English, extending this idea to resource-poor languages might be difficult due to the relatively small amount of anchors in their corresponding Wikipedia articles. Exploring other data resources to automatically construct large-scale pre-training data can remedy this issue. For example, given a word in the monolingual dictionaries, we can regard the word itself, the definition of this word, and the example sentence of this word as the MRC answer, query, and context respectively. We believe our MRC-style pre-training is still applicable for low-resource languages with such dictionaries.

**Comparison with Large Language Models**   In this paper, we did not compare PMR with large language models (LLM) for the following two reasons. First, existing MLMs are small in scale. Therefore, we are unable to find a suitable MLM to make a fair comparison with LLMs. Second, studies have shown that LLMs yield inferior results compared to smaller MLMs on span extraction tasks, particularly those involving structured prediction [41, 43, 60, 31]. Based on this fact, we mainly compare with existing strong generative methods of comparable model size.

**Few-shot NER results of SpanBERT**   We ran SpanBERT [20] in our NER few-shot settings. However, its performance was below our expectations. In all our few-shot settings, SpanBERT achieved an F1 score of 0 on CoNLL and WNUT datasets. Additionally, its performance on ACE04 and ACE05 datasets was significantly lower than RoBERTa [36]. Based on these outcomes, we only compare PMR with SpanBERT in the NER full-resource setting.

## A.2   Fine-tuning Tasks

For EQA, we use the MRQA benchmark [15], including SQuAD [45], TriviaQA [21], NaturalQuestion [25], NewQA [55], SearchQA [14], HotpotQA [66], BioASQ [56], DROP [13], DuoRC [48], RACE [26], RelationExtraction [28], TextbookQA [22]. EQA has always been treated as an MRC problem, where the question serves as the MRC query, and the passage containing the answers serves as the MRC context. For NER, We follow MRC-NER [32] to formulate NER into the MRC paradigm, where the entity label together with its description serves as the MRC query, and the input text serves as the MRC context. The goal is to extract the corresponding entities as answers. We use the Eq. 4 as the learning objective, where $Y_{i,j}^{ext}$ indicates that the input span $X_{i:j}$ is an answer/entity.

For sequence classification tasks, we construct the MRC query and context as followed. MCQA: The query is the concatenation of the question and one choice, and the context is the supporting document. MNLI: The query is the entailment label concatenated with the label description, and the context is the concatenation of the premise and hypothesis. SST-2: The query is the sentiment label concatenated with the label description, and the context is the input sentence. We use Eq. 3 to fine-tune the classification tasks. Note that only the correct query-context pair would get $Y^{cls} = 1$. Otherwise, the supervision is $Y^{cls} = 0$. During inference, we select the query-context pair with the highest $S_{1,1}$ among all MRC examples constructed for the sequence classification instance as the final prediction. We show concrete examples for each task in Table 7 and Table 8.

## A.3   Implementations

We download the 2022-01-01 dump[5] of English Wikipedia. For each article, we extract the plain text with anchors via WikiExtractor [3] and then preprocess it with NLTK [4] for sentence segmentation and tokenization. We consider the definition articles of entities that appear as anchors in at least 10 other articles to construct the query. Then, for each anchor entity, we pair its query from the definition article with 10 relevant contexts from other mention articles that explicitly mention the corresponding anchors and construct answerable MRC examples as described in Sec. 2. Unanswerable examples are formed by pairing the query with 10 irrelevant contexts.

---

[5]https://dumps.wikimedia.org/enwiki/latest

| Task | | Example Input | Example Output |
|---|---|---|---|
| **EQA** (SQuAD) | Ori. | Question: Which NFL team represented the NFC at Super Bowl 50? Context: Super Bowl 50 was an American football game to determine the champion of the National Football League (NFL) for the 2015 season. The American Football Conference (AFC) champion Denver Broncos defeated the National Football Conference (NFC) champion Carolina Panthers to earn their third Super Bowl title. | Answer: "Carolina Panthers" |
| | PMR | [CLS] Which NFL team represented the NFC at Super Bowl 50 ? [SEP] [SEP] Super Bowl 50 was an American football game to determine the champion of the National Football League (NFL) for the 2015 season . The American Football Conference (AFC) champion Denver Broncos defeated the National Football Conference (NFC) champion Carolina Panthers to earn their third Super Bowl title . [SEP] | (53,54) - "Carolina Panthers" |
| **NER** (CoNLL) | Ori. | Two goals in the last six minutes gave holders Japan an uninspiring 2-1 Asian Cup victory over Syria on Friday. | ("Japan", LOC); ("Syria", LOC); ("Asian Cup", MISC) |
| | PMR | [CLS] "ORG" . Organization entities are limited to named corporate, governmental, or other organizational entities. [SEP] [SEP] Two goals in the last six minutes gave holders Japan an uninspiring 2-1 Asian Cup victory over Syria on Friday . [SEP] | ∅ |
| | | [CLS] "PER" . Person entities are named persons or family . [SEP] [SEP] Two goals in the last six minutes gave holders Japan an uninspiring 2-1 Asian Cup victory over Syria on Friday . [SEP] | ∅ |
| | | [CLS] "LOC" . Location entities are the name of politically or geographically defined locations such as cities , countries . [SEP] [SEP] Two goals in the last six minutes gave holders Japan an uninspiring 2-1 Asian Cup victory over Syria on Friday . [SEP] | (32,32) - "Japan"; (40,40) - "Syria" |
| | | [CLS] "MISC" . Examples of miscellaneous entities include events , nationalities , products and works of art . [SEP] [SEP] Two goals in the last six minutes gave holders Japan an uninspiring 2-1 Asian Cup victory over Syria on Friday . [SEP] | (34,35) - "Asian Cup" |

Table 7: MRC examples of span extraction. Ori. indicates the original data format of these NLU tasks.

We use Huggingface's implementations of RoBERTa [62] as the MLM backbone. During the pre-training stage, the window size $W$ for choosing context sentences is set to 2 on both sides. We use the first $T = 1$ sentence as the MRC query. Sometimes, the sentence segmentation would wrongly segment a few words to form a sentence, which is not meaningful enough to serve as an MRC query. Therefore, we continue to include subsequent sentences to form the query as long the query length is short than 30 words. The learning rate is set to 1e-5, and the training batch size is set to 40 and 24 for PMR$_{base}$ and PMR$_{large}$ respectively in order to maximize the usage of the GPU memory. We follow the default learning rate schedule and dropout settings used in RoBERTa. We use AdamW [37] as our optimizer. We train both PMR$_{base}$ and PMR$_{large}$ for 3 epochs on 4 A100 GPU. Since the WAE is a discriminative objective, the pre-training is extremely efficient, which tasks 36 and 89 hours to finish all training processes for two model sizes respectively. We also reserve 1,000 definition articles to build a dev set (20,000 examples) for selecting the best checkpoint. Since the queries constructed by these definition articles have never been used in training, they can be used to estimate the general language understanding ability of the model instead of hand match. The hyper-parameters of PMR$_{large}$ on downstream NLU tasks can be found in Table 9 and Table 11 for full-supervision and few-shot settings respectively.

| Task | | Example Input | Example Output |
|---|---|---|---|
| **MCQA** (OBQA) | Ori. | Question: A positive effect of burning biofuel is: (A) shortage of crops for the food supply. (B) an increase in air pollution (C) powering the lights in a home. (D) deforestation in the amazon to make room for crops. Context: Biofuel is used to produce electricity by burning. | Answer Choice: C |
| | PMR | [CLS] A positive effect of burning biofuel is shortage of crops for the food supply . [SEP] [SEP] Biofuel is used to produce electricity by burning . [SEP] | ∅ |
| | | [CLS] A positive effect of burning biofuel is an increase in air pollution . [SEP] [SEP] Biofuel is used to produce electricity by burning . [SEP] | ∅ |
| | | [CLS] A positive effect of burning biofuel is powering the lights in a home . [SEP] [SEP] Biofuel is used to produce electricity by burning . [SEP] | (0,0) - "[CLS]" |
| | | [CLS] A positive effect of burning biofuel is deforestation in the amazon to make room for crops . [SEP] [SEP] Biofuel is used to produce electricity by burning . [SEP] | ∅ |
| **Sentence Classification** (SST-2) | Ori. | This is one of Polanski's best films. | Positive |
| | PMR | [CLS] Negative , feeling not good . [SEP] [SEP] This is one of Polanski 's best films . [SEP] | ∅ |
| | | [CLS] Positive , having a good feeling . [SEP] [SEP] This is one of Polanski 's best films . [SEP] | (0,0) - "[CLS]" |
| **Sen. Pair Classification** (MNLI) | Ori. | Hypothesis: You and your friends are not welcome here, said Severn. Premise: Severn said the people were not welcome there. | Entailment |
| | PMR | [CLS] Neutral. The hypothesis is a sentence with mostly the same lexical items as the premise but a different meaning . [SEP] [SEP] Hypothesis : You and your friends are not welcome here, said Severn . Premise : Severn said the people were not welcome there . [SEP] | ∅ |
| | | [CLS] Entailment . The hypothesis is a sentence with a similar meaning as the premise . [SEP] [SEP] Hypothesis : You and your friends are not welcome here, said Severn . Premise : Severn said the people were not welcome there . [SEP] | (0,0) - "[CLS]" |
| | | [CLS] Contradiction . The hypothesis is a sentence with a contradictory meaning to the premise . [SEP] [SEP] Hypothesis : You and your friends are not welcome here, said Severn . Premise : Severn said the people were not welcome there . [SEP] | ∅ |

Table 8: MRC examples of sequence classification.

## A.4 Analysis of Data Construction

In addition to the defaulted way of constructing MRC examples (the first sentence in the definition article is the query, and randomly find 10 contexts for pairing 10 MRC examples), we compare with some advanced strategies to pair the query and the context, including:

- Q-C Relevance: We still use the first sentence from the definition article as the query, but we only select the top P% or top P most similar contexts to the query, where the similarity score is computed as the combination of BM25 and SimCSE [17].

| Dataset | CoNLL03 | WNUT | ACE04 | ACE05 | MRQA | RACE | DREAM | MCTest | MNLI | SST-2 |
|---|---|---|---|---|---|---|---|---|---|---|
| Query Length | 32 | 32 | 64 | 64 | 64 | 128 | 128 | 128 | 64 | 64 |
| Input Length | 192 | 160 | 192 | 192 | 384 | 512 | 512 | 512 | 192 | 192 |
| Batch Size | 32 | 16 | 64 | 32 | 16 | 8 | 2 | 2 | 16 | 16 |
| Learning Rate | 2e-5 | 1e-5 | 2e-5 | 2e-5 | 2e-5 | 2e-5 | 2e-5 | 1e-5 | 1e-5 | 2e-5 |
| Epoch | 10 | 5 | 10 | 5 | 4 | 4 | 3 | 8 | 3 | 2 |

Table 9: Hyper-parameters settings in fine-tuning downstream tasks in full-supervision settings.

| ID | Strategy | Query | Context | CoNLL | SQuAD | DREAM | SST-2 |
|---|---|---|---|---|---|---|---|
| 0 | RoBERTa$_{base}$ | N.A. | N.A. | 92.3 | 91.2 | 66.4 | 95.0 |
| 1 | Random | First 1 | Random 10 | 93.2 | 92.2 | 66.7 | 94.8 |
| 2 | Q-C Relevance (top P%) | First 1 | top 30% | 93.0 | 91.9 | 65.5 | 95.3 |
| 3 | Q-C Relevance (top P) | First 1 | top 10 | 93.2 | 92.1 | 65.8 | 94.8 |
| 4 | Random (Defaulted) | First 1 | Random 10 + Unanswerable | 93.1 | 92.1 | 70.7 | 94.6 |
| 5 | Q-C Relevance (top P) | First 1 | top 10 + Unanswerable | 93.1 | 92.2 | 69.7 | 94.7 |
| 6 | Q Diversity | Random 5 | Random 10 + Unanswerable | 93.2 | 92.2 | 70.6 | 94.8 |
| 7 | C Diversity | First 1 | Cluster 10 + Unanswerable | 92.8 | 92.2 | 70.5 | 95.1 |

Table 10: We try various advanced strategies to pair the query and the context to form an MRC example. the **Query** and **Context** columns indicate how to select possible query and context for pairing. + Unanswerable indicates that PMR also uses Unanswerable examples and is also trained with $L_{cls}$. Models are base-sized.

- Q Diversity: In searching for an anchor, we hope the query should be diverse enough such that the model would not make a hard match between the fixed query and the anchor. Therefore, we randomly select one sentence from the first P sentences in the definition article to serve as the query for the anchor, while we keep the same context selection strategy.

- C Diversity: We hope the contexts should also be diverse enough such that they provide more possible usages of an anchor. Therefore, We use K-means[6] to cluster all contexts containing the anchor into P clusters and randomly select 1 context in each cluster. Similar scores in K-means are also obtained via SimCSE.

We compare those advanced strategies with our defaulted one in Table 10, where two span extraction and sequence classification tasks are selected for evaluating the effectiveness of these strategies. First, we make a fast evaluation with only $L_{ext}$ without unanswerable examples (i.e. Strategy 1,2,3). Comparing Q-C Relevance (top P%) against Q-C Relevance (top P), we can observe that it is better to sample contexts based on absolute values. In Wikipedia, the reference frequency of anchor entities is extremely unbalanced, where some frequent anchor entities such as "the United States" are referenced more than 200,000 times, while other rare anchor entities are only mentioned once or twice in other articles. Therefore, Q-C Relevance (top P%) would waste too much focus on the well-learned frequent anchor entities and affect the learning of other less frequent anchor entities.

Then, when trained on both answerable and unanswerable examples as well well guided with both $L_{cls}$ and $L_{ext}$, we only sample an absolute number of contexts. However, comparing among Strategy 4,5,6,7, no significant difference between these strategies and our random sampling is observed. We

---

[6]https://github.com/subhadarship/kmeans_pytorch

| Dataset | EQA | NER |
|---|---|---|
| Query Length | 64 | 32 |
| Input Length | 384 | 192 |
| Batch Size | 12 | 12 |
| Learning Rate | {5e-5,1e-4} | {5e-5,1e-4} |
| Max Epochs/Steps | 12/200 | 20/200 |

Table 11: Hyper-parameters settings in fine-tuning downstream tasks in few-shot settings.

| | F1 | EM |
|---|---|---|
| RoBERTa | 7.3 | 0.1 |
| T5-v1.1 | 12.6 | 0.0 |
| FewshotBART | 0.8 | 0.3 |
| PMR | 17.2 | 10.4 |

The Broncos took an early lead in Super Bowl 50 and never trailed. Newton was limited by Denver's defense, which sacked him seven times and forced him into three turnovers, including *a fumble* which they recovered for a touchdown. Denver linebacker Von Miller was named Super Bowl MVP, recording *five solo tackles*, 2½ sacks, and two forced fumbles.

1. How many solo tackles did Von Miller make at Super Bowl 50?
   *Gold*: five solo tackles
   *RoBERTa*: forced him into three turnovers, including ( ✗ )
   *T5-v1.1*: context: context: context: context: context: context: ( ✗ )
   *FewshotBART*: ∅
   *PMR*: five solo tackles ( ✓ )

2. Which Newton turnover resulted in seven points for Denver?
   *Gold*: a fumble
   *RoBERTa*: trailed. Newton was limited by Denver's defense, which sacked him seven times and forced him into three turnovers, including a fumble which they recovered ( ✗ )
   *T5-v1.1*: . context: Newton's first Super Bowl touchdown came in Super Bowl 50. context: ( ✗ )
   *FewshotBART*: Denver linebacker Von ( ✗ )
   *PMR*: two forced fumbles ( ✗ )

Figure 6: Zero-shot performance on SQuAD and a case study. The F1/EM scores are shown in the left-top corner.

suggest that the benefits from these heuristic strategies are marginal in the presence of large-scale training data. Therefore, in consideration of the implementation simplicity, we just use the Random strategy as our final PMR implementation.

## A.5 Zero-shot Learning

To reveal PMR's inherent capability from its MRC-style pretraining, we show its zero-shot performance in Figure 6, where the F1 and Exact Match (EM) scores on the entire SQuAD dev set and a case study in answering several questions are presented. Without any fine-tuning, our PMR achieves 10.4 EM, whereas T5 and RoBERTa can barely provide a meaningful answer, as shown by their near-zero EM scores. In the case study, our PMR correctly answers the first question. For the second question, although PMR gives an incorrect answer, the prediction is still a grammatical phrase. In contrast, RoBERTa and T5-v1.1 always perform random extractions and generations. Such a phenomenon verifies that PMR obtains a higher-level language digest capability from the MRC-style pretraining and can directly tackle downstream tasks to some extent.

## A.6 Fully-Resource Results

Table 12 compares PMR with strong approaches in full-resource settings. On EQA and NER, PMR can significantly and consistently outperform previous approaches, where $PMR_{large}$ achieves up to 3.7 and 2.6 F1 improvements over $RoBERTa_{large}$ on WNUT and SearchQA, respectively. For the base-sized models, the advantage of PMR is more obvious, i.e. 1.4 F1 over $RoBERTa_{base}$. Apart from those, we also observe that: (1) PMR can also exceed strong generative approaches (i.e. UIE, T5-v1.1) on most tasks, demonstrating that the MRC paradigm is more suitable to tackle NLU tasks. (2) RoBERTa-Post, which leverages our Wikipedia corpus (a subset of its original pre-training data) for MLM-style continued-pretraining, performs poorly on most tasks, especially those with natural-question queries (i.e. EQA and MCQA). (3) PMR can be applied on even larger MLM such as $ALBERT_{xxlarge}$ [27] to gain stronger representation capability and further improve the performance of downstream tasks. Such findings suggest that with our MRC data format and WAE objective, PMR can leverage the same data to learn a high level of language understanding ability, beyond language representation.

| EQA | Size | Unified | SQuAD | NewsQA | TriviaQA | SearchQA | HotpotQA | NQ | Avg. |
|---|---|---|---|---|---|---|---|---|---|
| RBT-Post$_{large}$ | 355M | ✗ | 93.0 | 70.9 | 80.9 | 86.8 | 79.8 | 79.9 | 81.9 |
| SpanBERT$_{large}$ [20] | 336M | ✗ | 93.1 | 72.3 | 78.1 | 83.2 | 80.9 | 82.3 | 81.7 |
| LUKE$_{large}$ [64] | 483M | ✗ | 94.5 | 72.1 | NA | NA | 81.9 | 83.3 | - |
| T5-v1.1$_{large}$ [44] | 800M | △ | 93.9 | 69.8 | 77.8 | 87.1 | 81.9 | 81.6 | 82.0 |
| RoBERTa$_{base}$ | 125M | ✗ | 91.2 | 69.0 | 79.3 | 85.0 | 77.9 | 79.7 | 80.4 |
| PMR$_{base}$ (OURS) | 125M | ✓ | 92.1 | 71.9 | 81.5 | 86.4 | 80.6 | 81.0 | 82.3 |
| RoBERTa$_{large}$ | 355M | ✗ | 94.2 | 73.8 | 85.1 | 85.7 | 81.6 | 83.3 | 84.0 |
| PMR$_{large}$ (OURS) | 355M | ✓ | 94.5 | 74.0 | 85.1 | 88.3 | 83.6 | 83.8 | 84.9 |
| ALBERT$_{xxlarge}$ | 223M | ✗ | 94.7 | 75.3 | 86.0 | 89.4 | 83.8 | 83.8 | 85.5 |
| PMR$_{xxlarge}$ (OURS) | 223M | ✓ | **95.0** | **75.4** | **86.7** | **89.6** | **84.5** | **84.8** | **86.0** |

| NER | Size | Unified | CoNLL | WNUT | ACE04 | ACE05 | Avg. |
|---|---|---|---|---|---|---|---|
| Roberta$_{large}$+Tagging [36] | 355M | ✗ | 92.4 | 55.4 | - | - | - |
| RBT-Post$_{large}$ | 355M | ✗ | 92.7 | 53.8 | 86.6 | 86.2 | 79.8 |
| SpanBERT$_{large}$ | 336M | ✗ | 90.3 | 47.2 | 86.4 | 85.4 | 77.3 |
| LUKE$_{large}$ [64] | 483M | ✗ | 92.4$^†$ | 55.2$^†$ | - | - | - |
| CL-KL$_{large}$ [59] | 550M | ✗ | 93.2$^†$ | 59.3$^†$ | - | - | - |
| BARTNER$_{large}$ [65] | 406M | ✗ | 93.2$^‡$ | - | 86.8$^‡$ | 84.7$^‡$ | - |
| T5-v1.1$_{large}$ [44] | 800M | ✓ | 90.5 | 46.7 | 83.9 | 82.8 | 76.0 |
| UIE$_{large}$ [38] | 800M | ✓ | 93.2♠ | 52.5 | 86.9♠ | 85.8♠ | 79.6 |
| RoBERTa$_{base}$ | 125M | ✗ | 92.3 | 53.9 | 85.8 | 85.2 | 79.3 |
| PMR$_{base}$ (OURS) | 125M | ✓ | 93.1 | 57.6 | 86.1 | 86.1 | 80.7 |
| RoBERTa$_{large}$ | 355M | ✗ | 92.6 | 57.1 | 86.3 | 87.0 | 80.8 |
| PMR$_{large}$ (OURS) | 355M | ✓ | **93.6** | **60.8** | 87.5 | 87.4 | **82.3** |
| ALBERT$_{xxlarge}$ | 223M | ✗ | 92.8 | 54.0 | 86.8 | 87.7 | 80.3 |
| PMR$_{xxlarge}$ (OURS) | 223M | ✓ | 93.2 | 58.3 | **88.4** | **87.9** | 82.0 |

Table 12: Performance on EQA (F1), and NER (F1). The best models are bolded. For EQA, as done in MRQA [15], we report the F1 on dev set and produce the results of SpanBERT and LUKE following the same protocol. Although we try hard to produce the results of LUKE for TriviaQA and SearchQA, its performance is unreasonably low. For CoNLL, we assume there is no additional context available and therefore we retrieve the results of CL-KL w/o context from [59]. Results labeled by $^†$, $^‡$, and ♠ are cited from [59, 65, 38], respectively.

