# OpenReview forum: "From Cloze to Comprehension: Retrofitting Pre-trained Masked Language Models to Pre-trained Machine Reader"
_NeurIPS.cc/2023/Conference — NeurIPS 2023 poster_

### Official Review · Reviewer_p962 · 2023-07-07

**Soundness:** 3 good
**Presentation:** 3 good
**Contribution:** 3 good
**Rating:** 5
**Confidence:** 4

**Summary:**


- Summary:

This paper presents a novel pretraining method for span-extraction tasks, such as machine reading comprehension and NER.
By appending labels to the inputs, classification tasks can also be solved in span-extraction way.

- Method:

This paper proposes to use the hyper links in wikipedia to construct a large scale dataset for span-extraction pretraining. By continue training a pretrained MLM model with the constructed data, the model can learn to extract entity spans given the entity description and the mention article.

- Experiments:

By pretraining RoBERTa with their constructed dataset, various machine reading comprehension tasks and NER tasks are improved.
Also, by applying their model to test classification tasks, their model can extract the supporting span for classification.


**Strengths:**

- The data for pretraining is collected unsupervisely, which make the data collection process more attractive to inspire other research work.
- The proposed method is straightforward and effective. The evaluation and results are comprehensive.

**Weaknesses:**

- The constructed dataset can also be used to finetune seq2seq model like T5 for span-extraction. But only MLM models are covered in this work.
- This paper claims their model support both span extraction and classification tasks. But there are less classification experiments in this submission. For example, full-resource results on classification.


**Questions:**

1. For an article with multiple different hyperlink anchors, which one is used when building the pretraining dataset?
2. Many downstream tasks are evaluated in this paper. Can you discuss the SOTA method/model for each of them? such as the advantage and disadvantage of the proposed method compared to these SOTA method/models?
3. Can you discuss if there is a domain issue of the proposed method? as the pretraining data is only from wikipedia.

**Limitations:**

limitations discussed in the appendix

---

> ### Author Rebuttal · Authors · 2023-08-10
>
> Thank you for carefully reading our manuscript. In the following, we provide a detailed response to every point you mentioned in the review.
>
> ***Weakness-1***: The constructed dataset can also be used to finetune seq2seq model like T5. But only MLM models are covered in this work.
>
> ***Response***: Yes, our constructed data can also be used to fine-tune the seq2seq model. We use the same dataset to further pretrain T5-v1.1 model,  referred to as T5-v1.1+PMR. Then we fine-tune the two models on CoNLL NER and SQuAD EQA. The table below shows that T5-v1.1 + PMR improves over T5-v1.1 in the full fine-tuning setting (-FT), where the entire training data is used for fine-tuning. This demonstrates the effectiveness of our data in enhancing seq2seq models. But it should be noticed that the fine-tuning results of T5-v1.1 + PMR trail behind those of PMR, despite the latter having a smaller model size. As discussed in our introduction, this can be attributed to the fact that the output space of seq2seq models is too complex for effectively learning the task-solving patterns from the training data of a discriminative task.
>
> |Method| #Params|CoNLL-FT| SQuAD-FT| CoNLL-0 | SQuAD-0 |
> |---|---|---|---|---|---|
> |T5-v1.1|220M|89.2|89.9|0|4.1|
> |T5-v1.1 + PMR|220M|89.6|91.3|13.7|28.4|
> |RoBERTa |149M |92.3 |91.2|0 |4.1 |
> | PMR|149M |93.1 | 92.1|4.2 | 11.4|
>
> Nevertheless, pre-training seq2seq models with our data bring other merits. For example, the above table also shows the results from the zero-shot setting (-0), where the model is directly evaluated on downstream tasks without any fine-tuning. These results indicate that our data can significantly improve the comprehension capability of seq2seq models with notably better zero-shot performance.
>
> ***Weakness-2***: This paper claims their model support both span extraction and classification tasks. But there are less classification experiments in this submission. For example, full-resource results on classification.
>
> ***Response***: As mentioned in our introduction, our primary goal is to better solve extraction tasks. The motivation for conducting classification experiments is to ascertain if our PMR method, originally developed for extraction, can benefit the classification tasks. Consequently, we perform fully fine-tuning experiments. The results reveal that PMR's fine-tuning performance is slightly superior to that of the baseline RoBERTa model, thereby demonstrating the effectiveness of our PMR approach in tackling classification tasks. Furthermore, we conduct an explainability study to highlight the additional benefits of PMR in enhancing the explainability of classification predictions. We would like to conduct more classification experiments to reinforce the contributions of our method in terms of addressing classification tasks.
>
> ***Question-1***: For an article with multiple different hyperlink anchors, which one is used when building the pretraining dataset?
>
> ***Response***: As mentioned in Section 3, we would randomly sample at most 10 anchors in the mention article to build the pre-training dataset.
>
> ***Question-2***: Many downstream tasks are evaluated in this paper. Can you discuss the SOTA method/model for each of them? such as the advantage and disadvantage?
>
> ***Response***: For EQA, the SOTA method is FewshotBART. It bridges the gaps between the pre-training and fine-tuning by reformulating EQA tasks as a masked token prediction problem.
>
> Pros:
> 1. FewshotBART is simple and does not require any pre-training.
> 2. FewshotBART is effective in an extremely few-shot setting. It surpasses our PMR on 16-shot EQA.
>
> Cons:
> 1. FewshotBART is not applicable for dealing with structured prediction tasks like NER.
> 2. As a generative method, FewshotBART is less effective than discriminative methods in learning extraction patterns from more training data. FewshotBART is lower than our PMR when at least 32 examples are provided. It is also significantly lower than another discriminative method, Splinter, under a 1024-shot setting.
>
> For NER, the SOTA method is UIE. It reformulates a structured prediction task as a text-to-text format, and leverages T5 to solve NER in this way.
>
> Pros:
> 1. UIE is effective and efficient for information extraction tasks.
>
> Cons:
> 1. UIE mainly emphasizes a structured prediction and is not applicable to more complex extraction tasks like EQA.
> 2. The input of UIE is some label words from target tasks. Therefore, UIE may learn to memorize the hard map between the label word and the target spans. In contrast, the input of PMR is a natural language query, which contains more comprehensive information about a task label and is more general. Due to this difference, UIE is less effective in domain transfer.
>
> ***Question-3***: Can you discuss if there is a domain issue of the proposed method?
>
> ***Response***: Yes, there might be some domain issues, including the difference in text style, text structure, and domain-specific knowledge. But these challenges are not significantly detrimental to the efficacy of our method for the following reasons:
> 1. As an encyclopedic data resource, Wikipedia encompasses knowledge from various domains, which can mitigate potential domain biases associated with our method.
> 2. Different from plain text data, the curated MRC data in our paper emphasize comprehension. This allows models to learn to thoroughly understand the interaction between the query and the context. The enhanced comprehension capability outweighs the potential drawbacks of overfitting to Wikipedia text. As demonstrated in Table 3, PMR exhibits improved OOD generalization capabilities compared to RoBERTa model and significantly outperforms a larger-sized T5 model.
> 3. Our data curation method is domain-agnostic and can be applied to domain-specific data containing hyperlinks, facilitating a more comprehensive understanding of target-domain knowledge.

---

> > ### Comment · Reviewer_p962 · 2023-08-21
> >
> > Thanks for your update!
> >
> > I appreciate the new results of T5+PMR.The main reason I hoped to see T5+PMR is because T5-large/T5-xl is stronger than RoBERTa. I am curious about whether PMR can improve such a strong model. However, in table above, a small version of T5 (t5-base) is presented.
> > My concern has been partially addressed.

---

> > > ### Author Response · Authors · 2023-08-21
> > > **Experiements on T5-large**
> > >
> > > ***Response:***:
> > > Thanks for your appreciation of our work. Given the limited discussion time, we are unable to pre-train T5-XL with our PMR data at this moment. But we would like to show some zero-shot experiments that we previously conducted on top of T5-large in the table below. Our PMR pre-training can lead to a significant improvements of 30.8 F1 and 33 Acc. on SQuAD and SST-2 repectively. This demonstrates that our PMR still has great potential in enhancing an even larger backbone model.
> > >
> > > |Method| #Params| SQuAD-0 | SST2-0|
> > > |---|---|---|---|
> > > |T5-v1.1|800M|7.2|50.6|
> > > |T5-v1.1 + PMR|800M|38.0|83.6|

---

### Official Review · Reviewer_cpnT · 2023-07-07

**Soundness:** 3 good
**Presentation:** 3 good
**Contribution:** 3 good
**Rating:** 6
**Confidence:** 5

**Summary:**

This work aims to retrofit pre-trained language models to pre-trained machine reading comprehension models by constructing training data from Wikipedia. This technique can alleviate the discrepancy between pre-trained models and downstream tasks, especially extraction-based tasks. Specifically, the anchors of Wikipedia articles are leveraged. The definition of the anchor entity is regarded as query while the surrounding text of the anchor is regarded as context. The PMR obtains state-of-the-art performance on several datasets of NER and EQA under few-shot setting.


**Strengths:**

1. The proposed pre-trained machine reader achieved state-of-the-art performance on few-shot NER task and few-shot Extractive Question Answering task. The retrofitted pre-trained machine reader has competitive OOD generalization ability as T5-v1.1.
2. Extensive experiments were conducted to prove the effectiveness of the retrofitting method. Also, the experiments on explainability were also interesting.
3. It is interesting to integrate the unanswerable examples into the pre-training.


**Weaknesses:**

There are no major weaknesses. However, there are several comments here.
1. I think that the comparison with fewshotBART is not that direct. Even though fewshotBART and PMR were both designed for bridging the gaps between pre-training and fine-tuning. However, the PMR leveraged extra training data (unlabeled) while fewshotBART improved the input-output format without further pre-training. So it seems that for extremely low-resource setting, the fewshowBART has more advantages (no need for extra training).
2. The idea is quite similar to Splinter. In the related work, it would improve the paper if the authors can better explain the differences. E.g. These two works both use the entity recurring for constructing the dataset. Difference: For Splinter, it leverages the recurring of the original paragraph, while for PMR, a synthetic paragraph (abstract of entity + original paragraph) is constructed.
3. It would be great if the model size of the existing work can be shown in the table 1, for a more fair comparison. E.g. in the fewshotBART paper, they used the base model (6 layers of encoder and 6 layers of decoder -- 12 layers in total) -- which should be compared to PMR_base model.


**Questions:**

Is simple string matching used for finding other mentions of anchor entity in text?


**Limitations:**

No limitation is discussed in this work. No potential negative societal impact.

---

> ### Author Rebuttal · Authors · 2023-08-10
>
> Thanks for your thorough reading of the manuscript. In the following, we provide a detailed response to every point you mentioned in the review.
>
> ***Weakness-1***: I think that the comparison with fewshotBART is not that direct. Even though fewshotBART and PMR were both designed for bridging the gaps between pre-training and fine-tuning. However, the PMR leveraged extra training data (unlabeled) while fewshotBART improved the input-output format without further pre-training. So it seems that for extremely low-resource setting, the fewshowBART has more advantages (no need for extra training).
>
> ***Response***: We understand your concerns about the comparison between FewshotBART and PMR. In fact, FewshotBART merely bridges the gap by adjusting the fine-tuning formulation for downstream tasks. In contrast, our PMR bridges this gap by exploring both the pre-training and fine-tuning sides. Specifically, for pre-training, we retrofit a pre-trained MLM model into a pre-trained MRC model. For fine-tuning, we adjust NLU tasks into the unified MRC formulation, which can be directly tackled via our PMR. In this sense, PMR and FewshotBART are not directly comparable as they approach the problem from different angles and levels. Therefore, the comparison between PMR and FewshotBART should be more appropriately explained as a comparison between PMR and BART when the pretraining-finetuning gaps are bridged. In fact, the MRC-style pre-training is efficient and we will release the PMR model (PMR-base, PMR-large) to the public in a manner similar to other pre-trained models. Consequently, it will be possible to directly fine-tune our PMR models on downstream tasks without the need to pre-train MLM models with MRC data each time.
>
> Regarding your concern about the extremely low-resource setting, it is true that PMR is less effective than FewshotBART in extremely low-resource settings (<16-shot), however, our PMR is clearly superior to FewshotBART in the more realistic low-resource settings (i.e., 32-shot, 128-shot and 1024-shot). In addition, PMR offers several other advantages over FewshotBART, such as robust full-resource performance, and explainability for classification. Consequently, we believe that PMR has the potential for a significant impact in the field of NLU.
>
> ***Weakness-2***: The idea is quite similar to Splinter. In the related work, it would improve the paper if the authors can better explain the differences. E.g. These two works both use the entity recurring for constructing the dataset. Difference: For Splinter, it leverages the recurring of the original paragraph, while for PMR, a synthetic paragraph (abstract of entity + original paragraph) is constructed.
>
> ***Response***: We appreciate your feedback and suggestions for enhancing our paper. In addition to the difference you mentioned, we have identified three more distinctions between PMR and Splinter:
> 1. Splinter is only limited to the QA domain, while PMR can be directly applied to a wide range of NLU tasks.
> 2. Splinter employs separate start and end predictors. During the inference process, it identifies the answer span with the highest sum of the probabilities of start and end positions. This means that Splinter's architecture is limited to single-answer detection and does not support multi-answer extraction. In contrast, our PMR employs a span-based predictor, allowing it to detect multiple spans in each inference time. Therefore, PMR can be used to solve NER tasks that require multi-span extraction.
> 3. The pre-training data of Splinter is grounded to the special token [QUESTION]. This may lead the model to memorize certain fixed patterns that could potentially hinder the generalization capability of the backbone models. In contrast, the pre-training data of PMR is grounded to a natural language query (abstract of entity).  As a result, a model trained with PMR data can acquire an enhanced comprehension capability of natural language, making it more generalizable compared to the model trained with Splinter data.
>
> ***Weakness-3***: It would be great if the model size of the existing work can be shown in the table 1, for a more fair comparison. E.g. in the fewshotBART paper, they used the base model (6 layers of encoder and 6 layers of decoder -- 12 layers in total) -- which should be compared to PMR_base model.
>
> ***Response***: Thanks for your suggestion to improve the presentation of our paper. We would like to indicate the model size in Table 1 and Table 2.
>
> ***Question***: Is simple string matching used for finding other mentions of anchor entity in text?
>
> ***Response***: Yes, your understanding is correct.
>
> ***Limitations***: No limitation is discussed in this work.
>
> ***Response***: Due to the page limitation, we put the limitation section in the appendix. However, we plan to move this section into the main text in the final version if an additional page is provided.

---

### Official Review · Reviewer_D3ec · 2023-07-17

**Soundness:** 3 good
**Presentation:** 4 excellent
**Contribution:** 3 good
**Rating:** 7
**Confidence:** 4

**Summary:**

The paper introduces PMR, a pretrained machine reader. PMR combines a MLM and extractor modules in an MRC-style using a synthetic dataset from Wikipedia in order to bring the gap between pretrain and finetuning for span extraction tasks. Experimental results in few shot settings across various benchmarks (EQA, NER, Sequence classification) highlight PMR's effectiveness compared to similarly sized MLM models. Furthermore, PMR's performance in out-of-distribution (OOD) scenarios is demonstrated through additional experiments.

**Strengths:**

- Motivations are clear and narrow their contributions to a specific field (Span extraction)
- The construction of a synthetic dataset from Wikipedia for MRC-style pretraining looks interesting.
- The proposed method is tested on several benchmarks.
- The paper is easy to read, and generally well-written.


**Weaknesses:**

- Authors decided not to include LLMs as baselines for generative finetuning approaches, however, there are smaller versions e.g. opt-350m, T5Flan-small, etc.
- The paper presents experiments on EQA for a medium-big number of shots. I would suggest adding results for 4 and 8 samples to verify if the number of few shot examples affects the performance of PMR in comparison to the other baselines.
- (Minor) It will be better if the authors mention limitations in the main section of the paper as is important for readers to quickly notice the limitations of the work.


**Questions:**

As mentioned in the weakness:
- I would suggest adding baselines for small versions of LLMs to make the paper more solid.
- Also include experiments in smaller few-shot settings for EQA benchmarks.
- (Minor) Include limitations in the main section of the paper.


**Limitations:**

Yes

---

> ### Author Rebuttal · Authors · 2023-08-10
>
> We thank you for your careful reading of the paper, positive feedback, and questions. Below we address specific questions raised by you.
>
> ***Weakness-1***: Authors decided not to include LLMs as baselines for generative finetuning approaches, however, there are smaller versions e.g. opt-350m, T5Flan-small, etc.
>
>
> ***Response***:  Thank you for your suggestions regarding the inclusion of smaller versions of LLMs. In fact, we have already incorporated some smaller LLMs such as UIE and T5-v1.1. We are open to including additional smaller LLMs to further strengthen our paper. Specifically, we would like to compare the performance of opt-350m and PMR-large, which have similar parameter sizes. As illustrated in the table below, opt-350m exhibits extremely poor performance across all SQuAD few-shot settings. These results may indicate that the smaller-sized LLM, opt-350m, is not as effective as PMR in addressing NLU tasks.
>
> |SQuAD| 16| 32|128|1024|
> |---|---|---|---|---|
> |opt-350m| $9.2_{0.7}$| $8.7_{0.8}$| $9.5_{1.3}$| $12.7_{1.3}$|
> |PMR-large (355m)| $60.3_{4.0}$| $70.0_{3.2}$| $81.7_{1.2}$| $87.6_{0.7}$|
>
>
>
>
> Regarding multi-task fine-tuned LLMs like T5Flan-small, it has access to more NLU  data than PMR and the baseline models. This gives T5Flan-small an advantage in terms of exposure to a diverse range of tasks and data, potentially improving its performance. Consequently, it may be not fair to compare T5Flan with our PMR and all of the baseline models.
>
>
> ***Weakness-2***: The paper presents experiments on EQA for a medium-big number of shots. I would suggest adding results for 4 and 8 samples to verify if the number of few shot examples affects the performance of PMR in comparison to the other baselines.
>
>
> ***Response***: For the EQA task, we utilize the few-shot datasets proposed by Splinter. The data has a minimum number of 16 shots. As a result, we are unable to conduct 4-shot or 8-shot experiments based on these datasets. However, it is possible to create smaller few-shot settings by us and perform experiments within these configurations.
>
>
> ***Weakness-3***: (Minor) It will be better if the authors mention limitations in the main section of the paper as is important for readers to quickly notice the limitations of the work.
>
>
> ***Response***: Thanks for your suggestion. Due to the page limitation, we put the limitation section in the appendix. However, we plan to move this section into the main text in the final version if an additional page is provided.

---

> > ### Comment · Reviewer_D3ec · 2023-08-21
> >
> > Thank you for addressing my concerns.
> > >Response: For the EQA task, we utilize the few-shot datasets proposed by Splinter. The data has a minimum number of 16 shots. As a result, we are unable to conduct 4-shot or 8-shot experiments based on these datasets. However, it is possible to create smaller few-shot settings by us and perform experiments within these configurations.
> >
> > I do understand that data has a minimum number of 16 shots but It would be great to show this in order to understand the real gain that comes from the model and increasing the number of shots.
> >
> > Overall, I believe the paper deserves to be accepted. In that sense, I maintain my score

---

### Official Review · Reviewer_bQaJ · 2023-07-27

**Soundness:** 3 good
**Presentation:** 3 good
**Contribution:** 3 good
**Rating:** 7
**Confidence:** 3

**Summary:**

The paper proposes Pre-trained Machine Reader (PMR) which allows for retrofitting MLMs for better transferability to downstream tasks, especially under low-resource scenarios. PMR introduces an MRC-style head on top of MLMs and is pre-trained with a large volume of MRC-style data constructed from Wikipedia anchors and hyperlinks. The pertaining task (Wiki Anchor Extraction (WAE)) is to determine if the context and the query are relevant, and if so answer extraction from the context. PMR is evaluated on span extraction tasks like named entity recognition (NER) and extractive question answering (EQA). It achieves significantly better performance than vanilla MLMs under low-resource settings.

**Strengths:**

The idea of using Wikipedia anchors to construct MRC-style data is interesting, and the paper is well-written. The results show that PMR performs significantly better than vanilla MLMs under low-resource settings.

**Weaknesses:**

It is unclear to me if the gains come from the WAE pretraining task itself or just having more MRC-style training data. The performance gains could stem from PMR having access to orders of magnitude more MRC examples during training with WAE rather than WAE specifically being a better task. Comparing WAE pretraining to an equivalent amount of pretraining data from existing MRC datasets would better isolate the impact of the pretraining task design and Wikipedia data.

**Questions:**

does WAE training return better performance on downstream tasks than training on other MRC tasks on the same amount of data ? I want to know this to see WAE pretraining task is a better task, or if gains could be matched by an equivalent amount of pretraining data using other tasks such as NER.

**Limitations:**

Please add a limitations section.

---

> ### Author Rebuttal · Authors · 2023-08-10
>
> We thank you for the very insightful feedback, comments, and constructive reviews. We have tried to address all of your remarks to the best of our abilities and think that this has helped us greatly improve the overall quality of our manuscript.
>
> ***Weaknesses***: It is unclear to me if the gains come from the WAE pretraining task itself or just having more MRC-style training data. The performance gains could stem from PMR having access to orders of magnitude more MRC examples during training with WAE rather than WAE specifically being a better task. Comparing WAE pretraining to an equivalent amount of pretraining data from existing MRC datasets would better isolate the impact of the pretraining task design and Wikipedia data.
>
> ***Response***: Thank you for your insightful suggestion to isolate the impact of the pretraining task design and Wikipedia data. Indeed, WAE is an MRC-style objective that is capable of jointly addressing extraction and classification tasks. WAE only functionally enables us to train an MRC model using hyperlinked data. Therefore, we conclude that the majority of the benefits stem from the extensive MRC pre-training data.
>
> In accordance with your recommendation, we conduct the following experiments: First, we trained a model with CoNLL NER data (RoBERTa + CoNLL), and second, we trained another model with our curated hyperlinked data of the same volume as CoNLL (RoBERTa + hyperlink-small). Both models were evaluated on SQuAD dev set to eliminate the influence of in-domain information. The results of these experiments can be found in the table below. Interestingly, utilizing our data for training with the WAE objective yields a slight improvement over the CoNLL data.
>
> |Method| #Params| SQuAD|
> |---|---|---|
> |RoBERTa + CoNLL |149M  |0 |
> |RoBERTa + hyperlink-small|149M | 5.7|
> |RoBERTa + SQuAD |149M |91.2 |
>
> However, the performance of these two models significantly lags behind the in-domain SQuAD training (RoBERTa + SQuAD), which may not provide sufficient evidence to draw a reliable conclusion. Therefore, we reformulate our hyperlinked data into a text-to-text format, subsequently using it to train a generative model with a seq2seq objective. As depicted in the table below, we employed the entire hyperlinked data to continuously train the T5-v1.1 model, reffered as T5-v1.1 + hyperlink-all. We then fine-tune these two models on downstream extraction tasks (CoNLL NER and SQuAD EQA). The T5-v1.1 + hyperlink-all model demonstrates improvement over the T5-v1.1 on both tasks. This observation is also consistent between the RoBERTa and PMR models presented in our paper. Based on these findings, our data appears to be generally effective in enhancing the comprehension capabilities of backbone models with different training objectives. This conclusion should verify our conjecture that the hyperlinked data accounts for the majority of the performance gains.
>
> |Method| #Params|CoNLL| SQuAD|
> |---|---|---|---|
> |T5-v1.1|220M|89.2| 89.9|
> |T5-v1.1 + hyperlink-all|220M|89.6|91.3|
> |RoBERTa |149M |92.3 |91.2 |
> | RoBERTa  + hyperlink-all (aka. PMR)|149M |93.1 | 92.1|
>
>
> ***Questions***: does WAE training return better performance on downstream tasks than training on other MRC tasks on the same amount of data ? I want to know this to see WAE pretraining task is a better task, or if gains could be matched by an equivalent amount of pretraining data using other tasks such as NER.
>
> ***Response***: Please see our response to the weakness part.
>
>
>
> ***Limitations***: Please add a limitations section.
>
>
> ***Response***: Thanks for your suggestion. Due to the page limitation, we put the limitation section in the appendix. However, we plan to move this section into the main text in the final version if an additional page is provided.

---

> > ### Comment · Reviewer_bQaJ · 2023-08-18
> >
> > Thank you Authors for your response. Regarding your second experiment comparing T5-v1.1 + hyperlink-all vs T5-v1.1 model. Is the T5-v1.1 model also continually trained with another MRC dataset of equal volume ? Similarly is RoBERTa also continually trained with another MRC dataset of equal volume?  If not, I'm not yet convinced that the gains are stemming from WAE pretraining task specifically or just the model seeing more data in general.

---

> > > ### Author Response · Authors · 2023-08-20
> > > **Effectiveness of WAE task**
> > >
> > > ***Response:*** Are you inquiring about models that undergo continual training with ***Wikipedia data of equal volume under the MLM pre-training task (i.e. the original pre-training task)*** rather than the WAE task? If so, we have already established this baseline on top of RoBERTa, as presented in Table 1, Table 2 (fewshot), and Table 12 (full-resource). This baseline is referred to as RBT-Post, where we further pre-train the RoBERTa model using the same volume of Wikipedia data but with an MLM objective. The comparison results between RBT-post and PMR suggest that the proposed WAE objective is still superior to the MLM objective when the model is additionally pre-trained on the same volume of data.
> > >
> > > To make this conclusion more convincing, we run another group of experiments on top of T5-v1.1.
> > >
> > > |T5-v1.1| #Params|CoNLL| SQuAD|
> > > |---|---|---|---|
> > > |T5-Vanilla|220M|89.2| 89.9|
> > > |T5-WAE|220M|89.6|91.3|
> > > |T5-Post |220M|89.0|89.7 |
> > >
> > > As can be seen, there are three variants listed in the above table:
> > > - T5-Vanilla: We do not conduct further pre-training but directly use the task-specific data to fine-tune the T5-v1.1.
> > > - T5-WAE: We continually pre-train T5-v1.1 using our MRC data constructed from Wikipedia under our WAE task.
> > > - T5-Post: We continually pre-train T5-v1.1 with its original loss on the Wikipedia texts that construct the WAE training set (they are of equal volume for sure).
> > >
> > > The comparison results between T5-WAE and T5-Post again demonstrate the effectiveness of the proposed WAE task. Therefore, apart from allowing the model to see more data, the WAE task itself indeed contributes to the performance gains.

---

> > > > ### Comment · Reviewer_bQaJ · 2023-08-22
> > > > **Clarification**
> > > >
> > > > I was inquiring about models that undergo continual training with similar data of equal volume not under the MLM pre-training task but another MRC-style task. This was partially addressed by the RoBERTa + CoNLL vs. the RoBERTa + CoNLL comparison. As such I maintain my score.

---

> > > > > ### Author Response · Authors · 2023-08-22
> > > > > **Effectiveness of WAE (PMR vs Splinter)**
> > > > >
> > > > > Thanks for your clarification. Table 1 presents a comparison between PMR and Splinter, which we believe can better address your concerns regarding the effectiveness of WAE (FYI, we also show the 16-shot EQA results from Table 1 below). Similar to our PMR method, Splinter is also an MRC-style continual pre-training method. Firstly, Splinter introduces a recurring span selection pre-training task. This task also involves predicting the start/end indexes of the answer, which could be regarded as an MRC-style pre-training task. Secondly, Splinter constructs its pre-training data from the entire English Wikipedia text, which is exactly of the same volume and from the same source as our WAE training data. Thirdly, both Splinter and PMR are initialized from RoBERTa. Hence, the comparison between Splinter and PMR can be viewed as a comparison between WAE and other MRC-style pre-training tasks (e.g. recurring span selection). The superior experimental results of PMR over Splinter suggest that WAE is indeed a better MRC-style pre-training task.
> > > > >
> > > > > |Method| #Params|SQuAD| TriviaQA|NQ| NewsQA| SearchQA| HotpotQA|BioASQ| TbQA|Avg.|
> > > > > |---|---|---|---|---|---|---|---|---|---|---|
> > > > > |Splinter |149M|54.6|18.9|27.4|20.8|26.3|24.0|28.2|19.4|27.5|
> > > > > |PMR |149M|46.5|47.7|32.6|26,2|50.1|32.9|49.1|27.9|39.1|

---

### Official Review · Reviewer_8871 · 2023-07-28

**Soundness:** 3 good
**Presentation:** 3 good
**Contribution:** 2 fair
**Rating:** 4
**Confidence:** 3

**Summary:**

To improve the performance of tasks that can be formulated as span extraction (e.g., extractive MRC and NER), especially in low-resource settings, this paper proposes to automatically construct data MRC-style data based on Wikipedia for model pre-training. Another advantage of this proposed method is that when applied to classification tasks, spans that can be used to explain the answer will be extracted for better explainability.






**Strengths:**

1) Several backbone models are used for comparison.

2) The authors show MRC-style formulation can address several types of NLP tasks.

**Weaknesses:**

1) The proposed work focuses on tasks that can be formatted as MRC tasks. The transformation may be not efficient (e.g., several follow-up questions may be asked and answered to arrive at the final solution) or effective for tasks that involve complex structures or a large number of types such as fine-grained NER, event extraction, and text generation tasks.

2) It may be useful to explore other types of models such as decoder-only and encoder-decoder models.

3) Evaluation: it might be better if the authors can compare the proposed method PMR with previous ones on these datasets. Are these baselines reported in Table 1 and Table 2 implemented by the authors? If not, references should be clearly stated. If the CoNLL refers to CoNLL-03, UIE only reports one/five/ten-shot results (46.43/67.09/73.9) on this dataset. This paper starts with four-shot experiments (65.7), making results not easily compared.

**Questions:**

Please check the above comments.

**Limitations:**

The proposed work focuses on tasks that can be easily formatted as MRC tasks.

---

> ### Author Rebuttal · Authors · 2023-08-10
>
> Thank you for the careful reading of the manuscript and the insightful remarks that will allow us to improve the quality of our work. Here are our answers to the identified weaknesses.
>
> ***Weakness-1***: The proposed work focuses on tasks that can be formatted as MRC tasks. The transformation may be not efficient (e.g., several follow-up questions may be asked and answered to arrive at the final solution) or effective for tasks that involve complex structures or a large number of types such as fine-grained NER, event extraction, and text generation tasks.
>
> ***Response***: As mentioned in our Introduction section, MRC has been proven to be a generalizable and effective paradigm for tackling various NLU tasks, including but not limited toNER [1], event extraction [2], text classification [3], regression [3] and coreference resolution[4]. For more complex NLU tasks, they can be broken down into multiple extraction and classification tasks, which can then be separately addressed using our PMR method. Therefore, PMR is fundamentally applicable for addressing all NLU tasks.
>
> In response to the efficiency concerns raised, we acknowledge that certain complex NLU tasks may be inefficient to tackle within the MRC paradigm. However, this should not be considered a severe problem. Several methods have been proposed to enhance the efficiency of MRC models, such as late fusion [5,6] and late interaction [7]. Additionally, we can incorporate a retrieve-then-read framework into the MRC model. This framework first retrieves a small set of the most relevant candidate task labels and subsequently concatenates the label-grounded query with the context for joint encoding.
>
> [1] A unified MRC framework for named entity recognition. Xiaoya Li, Jingrong Feng, Yuxian Meng, Qinghong Han, Fei Wu, and Jiwei Li. ACL2020.
>
> [2] Event extraction as machine reading comprehension. Jian Liu, Yubo Chen, Kang Liu, Wei Bi, and Xiaojiang Liu. EMNLP 2020.
>
> [3] Unifying Question Answering, text classification, and regression via span extraction. Nitish Shirish Keskar, Bryan McCann, Caiming Xiong, and Richard Socher.
>
> [4] CorefQA: Coreference resolution as query-based span prediction. Wei Wu, Fei Wang, Arianna Yuan, Fei Wu, and Jiwei Li. ACL2020.
>
> [5] Label Semantics for Few Shot Named Entity Recognition. Jie Ma, Miguel Ballesteros, Srikanth Doss, Rishita Anubhai, Sunil Mallya, Yaser Al-Onaizan, Dan Roth. ACL 2022 Findings.
>
> [6] Enhanced Language Representation with Label Knowledge for Span Extraction. Pan Yang, Xin Cong, Zhenyu Sun, Xingwu Liu. EMNLP 2021.
>
> [7] ColBERT: Efficient and Effective Passage Search via Contextualized Late Interaction over BERT. Omar Khattab, Matei Zaharia. SIGIR 2020.
>
> ***Weakness-2***: It may be useful to explore other types of models such as decoder-only and encoder-decoder models.
>
> ***Response***: Thanks for your suggestions. We would like to investigate the feasibility to apply our constructed data in generative models. Here we have some initial results in the table below:
>
> |Method| #Params|CoNLL-FT| SQuAD-FT| CoNLL-0 | SQuAD-0 |
> |---|---|---|---|---|---|
> |T5-v1.1|220M|89.2|89.9|0|4.1|
> |T5-v1.1w/ MRC-PT|220M|89.6|91.3|13.7|28.4|
>
> We conducted experiments using our MRC data to pre-train T5-v1.1 models, resulting in T5-v1.1 w/ MRC-PT. We then fine-tuned these four models on CoNLL NER and SQuAD EQA tasks. As can be seen, in the full fine-tuning setting (-FT), our MRC pre-training consistently improves the performance of a pre-trained encoder-decoder model (i.e., T5). Moreover, in the zero-shot setting (-0) where the model is directly evaluated on downstream tasks without any training, the model with our MRC pre-training can still achieve reasonable performance on these tasks while the vanilla T5 fails in almost all testing samples. As a result, our MRC-style pre-training has demonstrated efficacy across various model architectures.
>
> ***Weakness-3***: Evaluation: it might be better if the authors can compare the proposed method PMR with previous ones on these datasets. Are these baselines reported in Table 1 and Table 2 implemented by the authors? If not, references should be clearly stated. If the CoNLL refers to CoNLL-03, UIE only reports one/five/ten-shot results (46.43/67.09/73.9) on this dataset. This paper starts with four-shot experiments (65.7), making results not easily compared.
>
> ***Response***: Sorry for the confusion regarding the presentation of our results. In fact, we do compare our PMR with previous baselines on these datasets.  For EQA, we employ the few-shot datasets proposed by Splinter and compare our method with several strong baselines, including Splinter, FewshotBART, RoBERTa, and SpanBERT. The results of all four methods are directly cited from the Splinter paper.
>
> Regarding NER, since there is no consistent few-shot evaluation across different studies, we establish a new few-shot NER evaluation benchmark comprising four datasets with four few-shot settings for a fair and comprehensive comparison. We compare our method with strong few-shot NER baselines, including EntLM and UIE. All results are reproduced by us.
>
> To enhance the clarity of our paper, we will indicate the cited results and our reproduced results accordingly.

---

### Official Review · Reviewer_kcDM · 2023-08-01

**Soundness:** 3 good
**Presentation:** 3 good
**Contribution:** 3 good
**Rating:** 6
**Confidence:** 3

**Summary:**

The paper proposed a Pre-trained Machine Reader to close the gap the gap between pretraining and finetuning.

**Strengths:**

This paper presents a comprehensive and novel method to tackle the extractive tasks by pretraining a reader from curated large corpus from wikipedia. The methodology is sound to the specific task and the results are solid and strong.

**Weaknesses:**

This paper and the methodology mostly focus on the extractive tasks instead of generative tasks, limiting its scope and applicability.

**Questions:**

Why does the model generalize to unseen domains? Any insights or explaination?

---

> ### Author Rebuttal · Authors · 2023-08-10
>
> We thank you for the encouraging and insightful comments. Please find our responses to specific questions and concerns below.
>
> ***Weakness***: This paper and the methodology mostly focus on the extractive tasks instead of generative tasks, limiting its scope and applicability.
>
> ***Response***： Thank you for highlighting the scope and applicability concerns. To clarify, PMR primarily focuses on extractive tasks, while it is capable to tackle classification tasks as well. For an extraction task, PMR can be leveraged to detect the spans according to the question-grounded or label-grounded query, referred to Sec. 2.2. For a classification task with N labels, it can be regarded as  N relevance classification problems between each label (query) and the input text (context) within our PMR formulation, referred to Sec. 5.1. For more complex NLU tasks, they can be broken down into multiple extraction and classification tasks, which can then be effectively addressed using our PMR method. Therefore, PMR is fundamentally applicable for addressing all NLU tasks.
>
> Since the PMR framework is model-agnostic, it is easy to accomodate generative tasks by switching the foundation model from RoBERTa to the generation-oriented counterparts (e.g., T5 and OPT). To verify the effectiveness of our PMR on such generative-pretrained models, we reformulated the MRC-style hyperlinked data into a format of text-to-text generation and utilize the newly-formatted data to continuously pre-train a T5-v1.1 model, resulitng  in T5-v1.1+PMR. The table below presents the full fine-tuning (-FT) and zero-shot (-0) results These results demonstrate that our data effectively enhances the comprehension capabilities of a generative backbone model, leading to improved fine-tuning outcomes and remarkably strong zero-shot performance on CoNLL NER and SQuAD EQA tasks
>
> |Method| #Params|CoNLL-FT| SQuAD-FT| CoNLL-0 | SQuAD-0 |
> |---|---|---|---|---|---|
> |T5-v1.1|220M|89.2|89.9|0|4.1|
> |T5-v1.1+PMR|220M|89.6|91.3|13.7|28.4|
>
> ***Questions***: Why does the model generalize to unseen domains? Any insights or explaination?
>
> ***Response***: The success of generalization lies in two aspects. First, in our MRC formulation, the labels for each NLU task are represented as natural language queries rather than discrete values (as in a typical classification network). Natural language serves as a universally recognizable format for all NLP models. Consequently, MRC models can effectively generalize to unseen domains by comprehending queries grounded on unseen labels. Second, as an encyclopedic data resource, Wikipedia contains knowledge from various domains. The domain knowledge from Wikipedia also enhances the generalization capability of our pre-trained model. As a result, PMR can surpass RoBERTa on all six OOD EQA datasets with an +4.5 F1 on average.

---

### Decision · Program_Chairs · 2023-09-21

**Decision:**

Accept (poster)

**Comment:**

This paper proposes a clever idea for automatically constructing extractive QA data from Wikipedia. Pre-training a MLM on this data leads to improved performance, especially (unsurprisingly, given the lack of data) in lower-resource settings.

For this paper we had some issues getting the reviewers in place, leading to a larger number of reviews. The lowest score is also the lowest quality review and did not engage with the author response (I read the response, and think it alleviates most of the concerns). Overall, the majority of reviewers leans positive and I am inclined to agree.